# MultiScale Contextual Bandits for Long Term Objectives

**Richa Rastogi    Yuta Saito    Thorsten Joachims**
Department of Computer Science
Cornell University

## Abstract

The feedback that AI systems (e.g., recommender systems, chatbots) collect from user interactions is a crucial source of training data. While short-term feedback (e.g., clicks, engagement) is widely used for training, there is ample evidence that optimizing short-term feedback does not necessarily achieve the desired long-term objectives. Unfortunately, directly optimizing for long-term objectives is challenging, and we identify the disconnect in the timescales of short-term interventions (e.g., rankings) and the long-term feedback (e.g., user retention) as one of the key obstacles. To overcome this disconnect, we introduce the framework of MultiScale Policy Learning to contextually reconcile that AI systems need to act and optimize feedback at multiple interdependent timescales. Following a PAC-Bayes motivation, we show how the lower timescales with more plentiful data can provide a data-dependent hierarchical prior for faster learning at higher scales, where data is more scarce. As a result, the policies at all levels effectively optimize for the long-term. We instantiate the framework with **M**ulti**S**cale Off-Policy **B**andit **L**earning **(MSBL)** and demonstrate its effectiveness on three tasks relating to recommender and conversational systems.

## 1   Introduction

Many interactive AI systems (e.g., recommender systems, conversational systems) use abundantly collected short-term feedback for learning. However, it is well known that over-optimization of short-term feedback can adversely affect the long-term goals [14, 21, 26]. Similarly, challenges with competing time horizon objectives [40, 15], and issues like reward hacking [35, 29, 30] and user manipulation [18, 11] have been previously reported. For example, optimizing for engagement on social media platforms can lead to clickbait-y or toxic feeds. This neither reflects the user preferences nor the platform's goals of retaining users. We wish to design systems that optimize for long-term objectives, that are beneficial for various stakeholders in the system [1].

A key problem in achieving this goal is that the long-term feedback (e.g., user retention) is at a different timescale than the short-term interventions (e.g., rankings of recommended products) that are used to optimize them. For instance, optimizing rankings for clicks in a recommender system is relatively straightforward since clicks can be attributed to an individual ranking. Indeed, learning methods such as bandits that optimize for users' immediate response to recommendations such as clicks, likes, views, etc are widely used to learn ranking policies [19, 46]. Now, consider optimizing rankings for the feedback on users' subscription renewal, which is observed monthly. We note the disconnect between the timescale of rankings presented and the long-term objective of subscription renewal, making this a much harder problem than the previous one. While a Markov Decision Process can in principle model sequential dependencies to achieve long-term goals, in practice the large resulting state spaces, credit-assignment problems, and the sparsity of long-term feedback prohibit straightforward applications of reinforcement learning [23].

39th Conference on Neural Information Processing Systems (NeurIPS 2025).

To overcome this problem, our approach is to contextually reconcile the disconnect between short-term interventions and long-term objectives by learning interventions and policies at multiple timescales.

Consider a recommender system for a video streaming platform as depicted in Figure 1. At the lowest level, the system acts by providing rankings of recommended products. These rankings (short-term interventions) can be optimized for clicks (short-term reward). While clicks are an important signal for learning that is abundantly available at the lowest level of Figure 1, an unmitigated maximization of clicks is not necessarily aligned with higher-level goals. In particular, the platform may sacrifice some clicks if that leads to a higher weekly return rate. This metric is a more reliable indicator of user satisfaction, but it is available at a slower timescale. Finally, at the highest level, the platform ultimately cares about user retention and subscription renewal. This feedback lives at an even slower timescale, and it is thus more scarce but even more valuable. Similar multi-scale levels of feedback metrics also exist for other AI systems (e.g., tutoring chatbots that aim to achieve long-term learning outcomes), and additional settings are discussed in Appendix B.

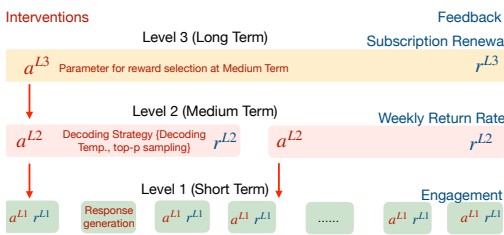

Figure 1: MultiScale feedback $r$ with corresponding interventions $a$ at each level $(L1, L2, L3)$. At the short-term level, engagement feedback (e.g., responses, clicks) is observed at the fastest timescale. At the next higher level, we observe feedback like the weekly return rate. And at an even higher level, subscription renewal is observed at the slowest timescale.

The main contributions of this work are as follows:

1. We introduce a new framework – **MultiScale Policy Learning –** that formalizes a multi-level approach for optimizing long-term objectives. The framework introduces a recursive construction of priors that are informed by data at the lower levels to speed up learning at the higher levels.

2. We demonstrate the practicality and generality of this framework by developing **MultiScale Off-Policy Bandit Learning (MSBL)** as a simple recursive algorithm for training multi-scale contextual bandit policies. It includes two widely applicable constructions for nested training of policies that enable the use of both abundant short-term and sparse long-term data for optimizing long-term outcomes.

3. We demonstrate the effectiveness of our approach empirically on three tasks ranging from recommender to conversational systems. Ablations show robustness of our method for optimizing the long term reward.

## 2   Related Works

**Hierarchical Reinforcement Learning.** Hierarchical RL (HRL) approaches, such as the options framework [37, 3] and feudal learning [7], learn to operate on different levels of temporal abstraction. HRL aims to accomplish complex tasks by dividing them into sub-goals with the higher level assigning sub-tasks to the next lower level, such as in robotic applications [31]. While our framework shares the idea of creating a hierarchy, the type of hierarchy is fundamentally different. In particular, our tasks do not have a subgoal decomposition as in HRL, where the discovery of the subgoals and their execution is critical and guides the micro policy. Instead, we exploit the hierarchy of feedback timescales to construct a hierarchical prior over the policy space to speed up learning for sparse long-term feedback. Our goal is to steer towards improving long-term outcomes, even at the expense of shorter-term rewards. This is fundamentally different from the macro actions (options) in HRL, which are abstractions over micro actions, and the goal is to combine these options as subroutines.

**Long Term Optimization.** There has been a growing interest in the study of recommender systems that go beyond optimizing engagement and clicks [25, 5, 23, 26, 6, 2]. Maystre et al. [23] provide an RL perspective to long-term learning in recommender systems and discuss challenges with credit attribution. Other approaches, such as [5] do not learn at the macro level and instead formulate long-term macro interventions that they aim to fulfill with minimum impact on short-term engagement metrics. These works operate on a single timescale and assume that the macro intervention is given and fixed. Our work can be viewed as learning the macro interventions themselves from long-term

feedback. In fact, these methods can be used for a single level-specific learning within our MultiScale framework, making them a special case of our approach.

**Multi-Objective Optimization.** Recent work has highlighted the importance of selecting weights for linear scalarization for multi-objective learning in recommender systems [27, 16, 42] and in text generation tasks [43]. A different line of work explores learning conditional policies as a single family of policies [9, 42, 43]. Our work leverages the idea of conditional policies and multi-objective learning from the perspective of optimizing for long-term outcomes. In doing so, we elevate single-stage learning to multiple levels with the macro level contextually selecting the objective to optimize at the micro level.

Additional related works are deferred to Appendix C.

## 3   Multi-Scale Policy Framework

We begin by providing a PAC-Bayesian motivation for why a hierarchical approach that exploits feedback across multiple scales can be substantially more data efficient. For simplicity of notation, we restrict to two levels – the micro level operating at the faster timescale $t1$ and the macro level operating at the slower timescale $t2$. As we will see later, the framework naturally extends to multiple scales of policy learning.

Our goal is to learn a policy $\pi(a|x)$ that selects an action $a$ for a given context $x$. We assume that contexts are drawn i.i.d. from an unknown distribution $x \sim p(x)$, but we conjecture that our approach can be extended to stateful models as well. Our approach rests on the realization that we observe rewards at different timescales in many AI systems. For two levels, we observe a reward

$$r^{L1} \sim p(r^{L1}|x,a)$$

at the micro level for each action $a$ and context $x$, corresponding to the short-term engagement (e.g., clicks). We also record rewards

$$r^{L2} \sim p(r^{L2}|(x_i, a_i), \ldots (x_{i+T}, a_{i+T}))$$

after we have taken a sequence of actions. This corresponds to the long-term feedback (e.g., weekly returns, subscription renewal) which we would like to optimize, since it typically better reflects stakeholder objectives. This leads us to the following policy learning objective,

$$\pi^{L2*} \leftarrow \arg\max_{\pi \in \Pi} V^{L2}(\pi), \quad \text{where} \quad V^{L2}(\pi) = \mathbb{E}_{x_i:x_{i+T}, a_i:a_{i+T}, r^{L2}}[r^{L2}]. \tag{1}$$

Unfortunately, for a large and complex policy space $\Pi$, finding the optimal policy $\pi^*$ by simply replacing the expected reward $V^{L2}(\pi)$ with its empirical estimate $\hat{V}^{L2}(\pi)$ on some training data $D^{L2}$ is typically intractable. The long-term reward is too infrequent for learning policies from scratch. As a result, existing approaches predominantly [44] optimize the more frequent reward signal $r^{L1}$ at the micro level.

$$\pi^{L1*} \leftarrow \arg\max_{\pi \in \Pi} V^{L1}(\pi), \quad \text{where} \quad V^{L1}(\pi) = \mathbb{E}_{x,a,r^{L1}}[r^{L1}]. \tag{2}$$

Note that even the policy $\pi^{L1*}$ that perfectly optimizes the micro level reward can have substantially worse reward at the macro level, $V^{L2}(\pi^{L1*}) < V^{L2}(\pi^{L2*})$. However, $\pi^{L1*}$ is typically much better than a random policy from $\Pi$. This raises the following question.

**How do we exploit feedback at the micro level to learn faster at the macro level?**   To provide a theoretical motivation, we make the following PAC-Bayesian argument. PAC Bayes generalization bounds [20, 24] provide uniform convergence over all posterior distributions $Q(\pi)$ for any given prior distribution $P(\pi)$. Specifically, with probability $1 - \delta$,

$$\left| \mathbb{E}_{\pi \sim Q}[V^{L2}(\pi)] - \mathbb{E}_{\pi \sim Q}[\hat{V}^{L2}(\pi; D)] \right| \leq O\left( \sqrt{\frac{KL(Q||P) + ln(1/\delta)}{n}} \right). \tag{3}$$

Since this bound holds for all $Q$, it also holds for any (approximately) optimal posterior $Q^{L2*}$ that maximizes the macro level reward. For a discrete policy space, $Q^{L2*}$ is the Dirac delta distribution centered on $\pi^{L2*}$. The bound states that this learning problem is 'easy' (i.e., requires a small number $n$ of training examples) if the KL-divergence between $Q^{L2*}$ and the prior $P$ is small.

**Can we improve the prior for the macro level with data from the micro level?** While the optimal policy $\pi^{L1*}$ at the micro level may be suboptimal at the macro level, a policy that is learned based on finite data at the micro level via $\hat{\pi}^{L1} \leftarrow \arg\max_{\pi \in \Pi} \hat{V}^{L1}(\pi)$ can provide useful prior information for learning at the macro level.

Consider the following illustrative example, where each policy $\pi(.|.,\theta) \in \Pi$ is defined via a parameter vector $\theta$. We denote $\theta^{L1}$ as the parameters of policy $\hat{\pi}^{L1}$, and $\theta^{L2*}$ as those of the optimal macro policy $\pi^{L2*}$. For simplicity of demonstration, we define the target policy distribution as $Q^{L2*} = N(\theta^{L2*}, \Sigma^{L2})$, an uninformed prior distribution $P_0 = N(\theta_0, \Sigma_0)$ for some arbitrary $\theta_0$, and an informed prior $P^{L1} = N(\theta^{L1}, \Sigma^{L1})$ centered at the learned micro policy $\hat{\pi}^{L1}$. The difference in training samples $n_0 - n^{L2}$ to get the same confidence interval for $Q^{L2*}$ in Equation (3) is proportional to $KL(Q^{L2*}||P_0) - KL(Q^{L2*}||P^{L1})$. We show in Section E.1 that for an appropriately chosen $\Sigma^{L1}$, the improvement in required training samples by moving to the informed prior $P^{L1}$ is at least

$$n_0 - n^{L2} \propto KL(Q^{L2*}||P_0) - KL(Q^{L2*}||P^{L1}) \in O(|\theta^{L2*} - \theta_0|_M - |\theta^{L2*} - \theta^{L1}|_M) \quad (4)$$

where $|\theta^{L2*} - \theta_0|_M$ is the squared Mahalanobis distance in the parameter space $\theta$. This shows that the policy $\hat{\pi}^{L1} \sim P^{L1}$ learned at the micro level can provide training sample savings at the macro level, if $\hat{\pi}^{L1}$ and $\pi^{L2*}$ are close compared to the uninformed prior $P_0$. In particular, if we can learn almost all parameters of $\pi^{L2*}$ at the micro level, the distance $|\theta^{L2*} - \theta^{L1}|_M$ will be small, resulting in higher macro level sample savings. The figure on the right illustrates how an informed prior $P^{L1}$ pulls the center of the prior $P_0$ closer to the parameters of $\pi^{L2*} \sim Q^{L2*}$.

To quantify the reduction in macro level training data, we construct a numerical example (detailed in Section E.1) with $\theta \in \mathbb{R}^{50}$, such that 49 parameters are learned well at the micro level and only 1 parameter needs to be adjusted at the macro level. This results in saving $\approx 98\%$ training samples[1], which means gathering enough macro level data may only take weeks instead of years.

# 4 Multi-Scale Policy Learning

In order to put the theoretical insights from the previous section into a practical policy learning algorithm, we introduce a factorization of contexts and policies at each level. We propose the policy factorization in a way that the micro level learns a large part of the parameter space, even if it is not aligned with the long term expected reward. This simplifies learning at macro level compared to learning from scratch.

**Multi-Scale Contexts.** The context $x \sim p(x)$ can be factorized as $p(x) \triangleq p(x^{L2}) \cdot p(x^{L1}|x^{L2})$. At the upper level, contexts $x^{L2} \sim p(x^{L2})$ arrive at timescale $t2$. An example of an upper-level context could be a user as described by demographic features or some long-term profile. For each such upper-level context, a sequence of lower-level contexts $x^{L1} \sim p(x^{L1}|x^{L2})$ is drawn conditionally on $x^{L2}$. Such lower-level contexts could be search queries, or chat requests.

**Multi-Scale Policies.** We consider the following factorization of policy space $\Pi$,

$$\Pi \triangleq \Pi^{L1} \cdot \Pi^{L2}, \quad (5)$$

where $\Pi^{L1}$ consists of micro policies, that as we will see later, can provide a strong inductive bias for the long term optimal policy. In contrast, the macro level is only concerned with learning within the space of reasonable policies obtained after micro learning and has a much smaller policy space $\Pi^{L2}$.

Specifically, for each upper-level context $x^{L2}$, the upper-level policy $\pi^{L2}$ selects an action $a^{L2}$ from action space $\mathcal{A}^{L2}$. Examples of $a^{L2}$ are diversity boosts, aggressiveness of spam filtering, the decoding strategy of an LLM policy, etc.

$$a^{L2} \sim \pi^{L2}(a^{L2}|x^{L2}) \quad ; \quad \pi^{L2} \in \Pi^{L2}$$

Importantly, as shown in Figure 2 (a), the action $a^{L2}$ corresponds to a lower-level policy $\pi_{a^{L2}}^{L1}$. This means that the action space $\mathcal{A}^{L2}$ at the upper level is isomorphic to a family of policies $\hat{\Pi}^{L1} = \{\hat{\pi}_{a^{L2}}^{L1} : a^{L2} \in \mathcal{A}^{L2}\}$, learned empirically at the lower level.

$$\mathcal{A}^{L2} \cong \hat{\Pi}^{L1}$$

---

[1]This calculation does not consider the investment of training samples $n^{L1}$ for constructing $P^{L1}$. Generally, these $n^{L1}$ samples are significantly cheaper than the macro samples $n^{L2}$, since they are $T$ times more frequent.

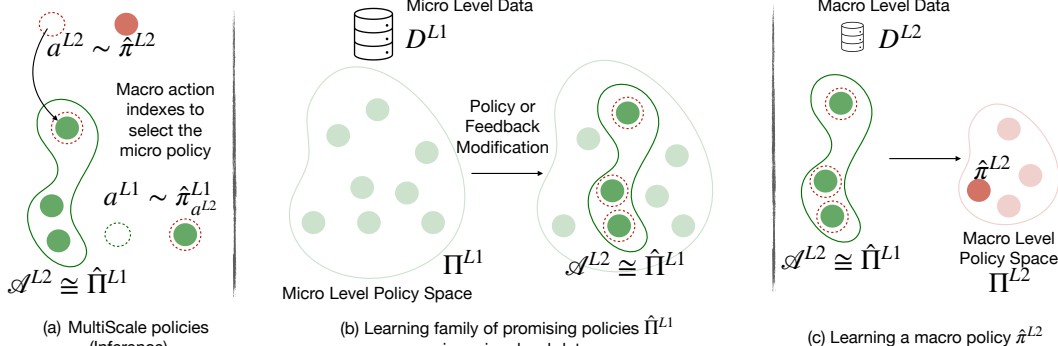

(a) MultiScale policies
(Inference)

(b) Learning family of promising policies $\hat{\Pi}^{L1}$
using micro level data

(c) Learning a macro policy $\hat{\pi}^{L2}$

Figure 2: (a) At inference, a macro action indexes to select the particular micro policy from a family of micro policies. The macro action space $\mathcal{A}^{L2}$ is isomorphic to the family of policies $\hat{\Pi}^{L1}$ (b) Learning micro policies: Abundant micro-level data is used to learn promising policies $\hat{\Pi}^{L1}$ using policy or feedback modification (c) Macro-level data is used to learn a macro policy. For more than two levels, (b) and (c) are recursively called, narrowing down micro policy space/ macro action space.

The lower-level policy $\pi_{a^{L2}}^{L1}$ indexed by the chosen upper-level action $a^{L2}$ will be executed for the subsequent lower-level contexts.

$$a^{L1} \sim \pi_{a^{L2}}^{L1}(a^{L1}|x^{L1})$$

The actions $a^{L1} \in \mathcal{A}^{L1}$ at the lower level are rankings, chat responses, or push notifications, just like in the conventional contextual bandit framework.

**Multi-Scale Data.** We collect contextual bandit feedback at both the upper and the lower level. At the lower level, we get the conventional data

$$D^{L1} = \{(x_i^{L1}, a_i^{L1}, r_i^{L1}, p_i^{L1})\}_{i=1}^{n^{L1}},$$

where we use the logging policy $\pi_0^{L1}$ and record the propensity $p_i^{L1} = \pi_0^{L1}(a_i^{L1}|x_i^{L1})$ to enable off-policy learning. However, unlike for conventional bandit policies, we also get data for the upper-level, which includes the long-term rewards

$$D^{L2} = \{(x_i^{L2}, \pi_{a_i^{L2}}^{L1}, r_i^{L2}, p_i^{L2})\}_{i=1}^{n^{L2}} \quad \text{with propensity} \quad p_i^{L2} = \pi_0^{L2}(a_i^{L2}|x_i^{L2}).$$

We thus need to devise an algorithm for learning both $\hat{\pi}^{L2}$, as well as each of the policies in $\hat{\Pi}^{L1}$. We approach this problem by using the upper-level data $D^{L2}$ for learning $a^{L2}$, and we use the lower-level data $D^{L1}$ to learn a comparably small set of policies $\hat{\Pi}^{L1}$ that serve as actions for the upper-level. This provides the affordance to use the abundant feedback in $D^{L1}$ for learning in the large action space $\mathcal{A}^{L1}$, thus narrowing down the set of potential actions for learning $\pi^{L2}$ from the comparably scarce data in $D^{L2}$, as shown in Figure 2 (b) and (c).

## 4.1 Learning a Family of Micro Policies

The following proposes two options, which both define a comparably small $\hat{\Pi}^{L1}$ based on the more abundant data $D^{L1}$ available at the lower level.

**Policy Modification.** In this procedure, we first learn a single policy $\pi^{L1}$ from $D^{L1}$ that we then modify to define the family $\hat{\Pi}^{L1}$. In particular, we first train $\hat{\pi}^{L1}$ to optimize the expected reward at the lower level according to Eq. (2). This single policy is then modified by each action $a^{L2}$, where each action takes the form of a function from $\Pi^{L1} \rightarrow \hat{\Pi}^{L1}$.

$$\hat{\pi}_{a^{L2}}^{L1} := a^{L2}(\hat{\pi}^{L1}) \tag{6}$$

In the example of text generation, a decoding strategy $a^{L2}$ might modify the learned LLM policy for more varied response generations. Similarly, applying a boost $a^{L2}$ to items of a particular type in a recommender system updates the ranking policy $\pi_{a^{L2}}^{L1}$ by ranking boosted items higher up, or items suspected to be click-bait lower. As a result, Eq. (6) defines policy update $\hat{\pi}_{a^{L2}}^{L1}$ for a given

intervention $a^{L2}$ as a form of perturbing the short-term optimized policy $\hat{\pi}^{L1}$. While $\hat{\pi}^{L1}_{a^{L2}}$ could provide a lower expected short-term reward as compared to that from $\hat{\pi}^{L1}$, it can be more effective at optimizing the reward at the upper-level (e.g., fewer clicks by more aggressively pruning suspected click bait can lead to better weekly returns).

**Feedback Modification.** In this procedure, each upper-level action $a^{L2}$ takes a form of a loss function that acts upon the observed feedback at the lower level when learning the lower-level policy. In this case we assume that the feedback $r^{L1}$ is vectorial (e.g., clicks, likes, purchases, add-to-carts), and each $a^{L2}$ is a different function (e.g., convex combination) for combining the feedback vector into a scalar loss. Then, for any given $a^{L2}$, we optimize

$$\pi^{L1}_{a^{L2}} := \arg\max_{\pi^{L1} \in \Pi^{L1}} \mathbb{E}_{p(x^{L1}), \pi^{L1}(a^{L1}|x^{L1}), p(r^{L1}|x^{L1}, a^{L1})}[a^{L2}(r^{L1})] \tag{7}$$

to get a family of policies $\hat{\Pi}^{L1}$.

For a more efficient implementation that does not require us to explicitly enumerate the policies in $\hat{\Pi}^{L1}$, we include $a^{L2}$ in the context and parameterize the reward $a^{L2}(r^{L1})$ during training $\pi^{L1}$ for every $a^{L2} \in \mathcal{A}^{L2}$. In this way, we only learn a single policy that is parameterized by $a^{L2}$ to represent all policies in $\hat{\Pi}^{L1}$. At inference, $a^{L2}$ chosen from the learned macro policy is included in the context of the micro policy, selecting the particular $\pi^{L1}_{a^{L2}}$. As a result, while Eq. (7) in principle refers to a discrete set of policies, practically we only train one micro policy. For LLM policies, this can be implemented as described in [43].

Note again that the transformation of the rewards can lead to lower short-term reward on some primary metric (e.g., clicks), but that the upper-level policy now has a space of actions that can optimize the longer-term metric (e.g., user retention).

## 4.2 Learning the Macro Policy

Once the family of policies $\hat{\Pi}^{L1}$ that correspond to the upper-level actions $a^{L2} \in \mathcal{A}^{L2}$ is fixed, we can chose from a wide range of policy-learning methods that use the data in $D^{L2}$ to optimize the expected upper-level reward $V^{L2}(\pi^{L2})$. In particular, we can use off-policy policy-gradient methods that optimize the inverse propensity weighted empirical average

$$\hat{\pi}^{L2} = \arg\max_{\pi^{L2}(.|., \theta)} \frac{1}{n^{L2}} \sum_{i=1}^{n^{L2}} \frac{\pi^{L2}(a_i^{L2}|x_i^{L2}, \theta)}{p_i^{L2}} r_i^{L2} \tag{8}$$

as an estimate of the expected upper-level reward $V^{L2}(\pi^{L2})$. If the policy $\hat{\pi}^{L2}(.|., \theta)$ is differentiable in its parameters $\theta$, we can use stochastic-gradient descent for training [17].

## 4.3 MultiScale Bandit Learning Algorithm

We can now summarize our approach to learning a nested set of policies across multiple levels in Algorithm 1. The algorithm uses off-policy contextual bandits [10, 38, 33] at each level. Algorithm 1 is limited to two levels for conciseness of notation, but the full recursive procedure for an arbitrary number of levels is given in Appendix D.2. In the experiments, we will explore policy spaces with two and three levels.

---

**Algorithm 1** MultiScale Training: Off-Policy Contextual Bandits

---

**Procedure** *PolicyLearning($\pi_0^{L2}, \pi_0^{L1}$)*

    Collect Micro Logged dataset $D^{L1} := \{(x_i^{L1}, a_i^{L1}, r_i^{L1}, p_i^{L1})\}_{i=1}^{n^{L1}} \sim \pi_0^{L1}$

    Learn Micro policies $\hat{\Pi}^{L1}$(Eq. (6) or (7) using $D^{L1}$)

    Collect Macro Logged dataset $D^{L2} := \{(x_j^{L2}, a_j^{L2}, r_j^{L2}, p_j^{L2})\}_{j=1}^{n^{L2}} \sim \pi_0^{L2}$

    Learn Macro Policy $\hat{\pi}^{L2} \leftarrow \arg\max_{\pi^{L2}} \hat{V}^{L2}(\pi^{L2}; D^{L2})$ (Eq. (8))

    **return** learned policies $\hat{\pi}^{L2}, \hat{\Pi}^{L1}$

---

The procedure requires the logging policies $\pi_0^{L2}, \pi_0^{L1}$ as input. We first collect logged bandit data $D^{L1}$ and learn the micro policies either as policy or feedback modification to get a family of policies

$\hat{\Pi}^{L1} := \{\hat{\pi}_{a^{L2}}^{L1} : a^{L2} \in \mathcal{A}^{L2}\}$. With the learned policies $\hat{\Pi}^{L1}$, we now collect logged bandit data $D^{L2}$ for the macro level. Note that the logged datasets $D^{L1}$ and $D^{L2}$ can be collected asynchronously, because during micro policy learning, we learn $\hat{\Pi}^{L1}$ for all $a^{L2} \in \mathcal{A}^{L2}$.

Overall, at each lower level $L(k-1)$, we use data $D^{L(k-1)}$ to learn the promising policies $\hat{\Pi}^{L(k-1)}$ that provide prior information and serve as the action space for the next upper level $L(k)$. While the training involves bottom-up learning of the policies, deployment involves top-down inference from learned policies at each of the levels. Figure 2 (a) shows the inference with actions chosen from the upper-most level policy, indexing the next lower level policy. For e.g., $a^{L2}$ is first selected from the learned $\hat{\pi}^{L2}(.|x^{L2})$ and the lower level action $a^{L1}$ is selected according to the indexed policy $\hat{\pi}_{a^{L2}}^{L1}(.|x^{L1})$. In this way, learning a family of policies enables adaptation (according to the intervention $a^{L2}$) for the micro level at inference time. We provide a formal inference algorithm in Section D.

Since we use off-policy learning independently at each level, convergence for the contextual bandit learning at each level depends on the number of samples and the action space [10]. As a result, with a smaller macro action space $\mathcal{A}^{L2}$ compared to $\mathcal{A}^{L1}$, we can utilize the scarcer data at the upper level and get similar convergence as the micro policy that utilizes the more abundant data.

## 5 Experiments

We conduct experiments on three scenarios related to conversational systems (Anthropic Helpful Assistant [4] and a new simulator), and a conventional recommendation system (KuaiRand video streaming benchmark) for two and three timescale levels. We provide an additional experiment on a toy domain that is simple enough to make RL tractable in Section F.1, and as expected, we find that our approach is competitive. In the following experiments, we compare our approach against single-stage policies that cannot personalize at the macro level, a random baseline policy (denoted by $a^{L2} \sim \pi_0^{L2}$) that selects interventions uniformly, and an oracle skyline policy. We use subscript index to denote the static policies, e.g., $a_1^{L2}$ is fixed action policy that always applies first macro intervention. Complete experiment setup, hyperparameters, and training details are provided in Section F.

### 5.1 Multi-turn Conversation

Figure 3 (a) shows a two level setup for multi-turn conversations. In this task, we learn the preference weight vector $a^{L2}$ for harmlessness and helpfulness with a bandit policy $\pi^{L2}(a^{L2}|x^{L2})$. The macro intervention $a^{L2}$ is applied as a *feedback modification* to the lower level LLM policy. The upper-level context $x^{L2}$ represents a user persona, such as "Child" and "Expert". At the micro level, the context $x^{L1}$ starts from a question in the Anthropic dataset. For each subsequent turn, the trained LLM policy $\hat{\Pi}^{L1}$ responds, and the user persona LLM asks a follow-up question. The short term observed reward is the user LLM's evaluation of a single-turn. We simu-

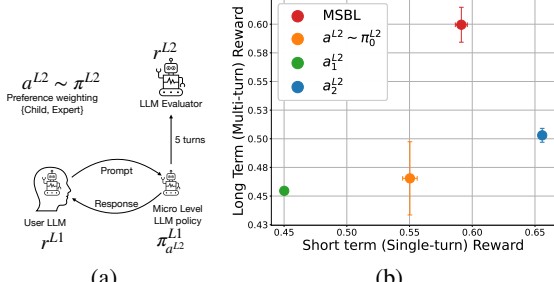

(a)                                    (b)

Figure 3: Multi-turn conversation: (a) Setup for learning preference weights $a^{L2}$ using **feedback modification** (b) Comparison of long-term (user satisfaction of multi-turn) vs short-term (single-turn) rewards for all users across 5 random seeds.

late five turns of the prompt-response cycle. At the end of the five turns, an LLM evaluator at the macro level scores user satisfaction for the given user persona and full conversation in $\{0, 1\}$.

**Results.** Figure 3 (b) shows that our approach of learning a macro level policy that selects the helpful/harmless tradeoff $a^{L2}$ for each user persona based on macro level feedback achieves the best long term reward for the overall conversation as compared to the non-adaptive baselines. Optimizing only the per-turn response can adversely affect the overall conversation. This is due to harm-inducing responses, which provide acceptable individual answers at the lower level but adversely affect the conversation over five turns. This experiment demonstrates that macro level learning is crucial and that MSBL effectively learns to optimize for long-term reward even when the reward is highly non-linear (i.e., Llama-3-70b [13] evaluations). Further results are given in Section F.4.

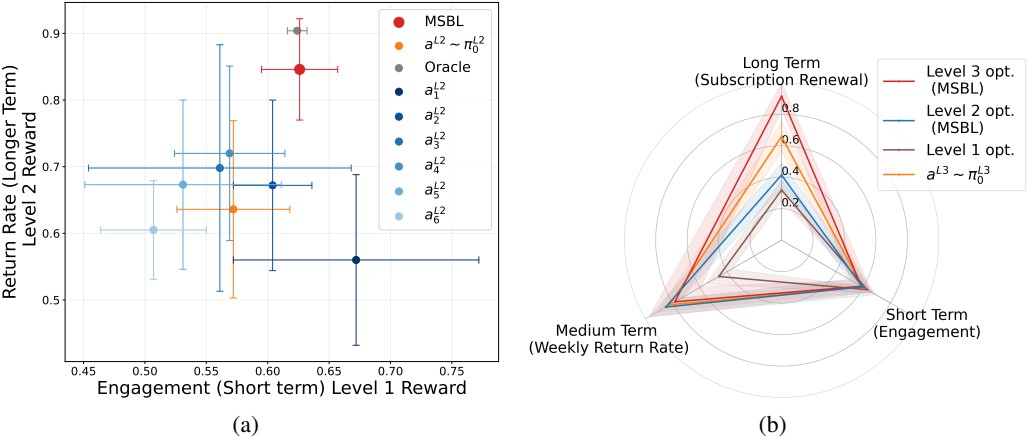

(a)                           (b)

Figure 4: Conversational recommender system: (a) Tradeoff between longer-term Level 2 and short-term Level 1 rewards using decoding temperature $a^{L2} \in \{0.0, 0.2, 0.4, 0.6, 0.8, 1.0\}$ as **policy modification**. (b) Tradeoff between expected rewards at all three levels. Expected rewards are reported across 5 random seeds for all users.

## 5.2 Conversational recommender system

To increase the complexity of the experiments and evaluate more than two levels of feedback, we built a simulated conversational recommender system based on the motivating example in Figure 1. We simulate 1500 users for training and test 300 users. At the lowest level, we use a pretrained LLM policy $\pi^{LLM}$ that acts as a personalized agent, generating cuisine suggestions $y$ to users at L1 timescale $t1 = \{1, \ldots 10\}$. Each query consists of a system prompt specifying the agent's expertise and a user query $q$. We learn a bandit policy $\pi^{L1}(a^{L1}|x^{L1})$ that selects the particular LLM agent $a^{L1}$ according to user context $x^{L1}$. Next, we generate response $y_t \sim \pi^{LLM}(.|a^{L1}, q, y_{t-1})$ from the LLM policy, given the agent selection $a^{L1}$, and append the previous timestep response $y_{t-1}$ to the current query $q$. The L1 reward is the inverse perplexity conditional on the optimal action, which simulates the engagement (relevance) metric. At the second level, we simulate two user groups with unknown preferences for relevance and diversity. L2 reward is a non-linear function of relevance and diversity, representing weekly return rate at every 10 timesteps of L1. The L2 policy $\pi^{L2}$ selects the decoding temperature $a^{L2} \in \{0.0, 0.2, 0.4, 0.6, 0.8, 1.0\}$ as a *policy modification* to the L1 policy. At the third level, we simulate two user groups, with different long term preferences and use *feedback modification* for L2 training to get a family of policies $\hat{\Pi}^{L2} := \{\hat{\pi}_{a^{L3}}^{L2} : a^{L3} \in \mathcal{A}^{L3}\}$.

**Results with two levels.** Figure 4 (a) illustrates the tradeoff between Level 1 and Level 2 expected rewards. While greedy decoding with $a_1^{L2}$ provides the best overall short term reward, it leads to low longer term reward at Level 2. Stochastic decoding temperatures $a_2^{L2}$ through $a_6^{L2}$ applied to the lower level policy improve the longer term reward depending on the user group but fall short. MSBL learns nested contextual policies $\hat{\pi}^{L1}$ and $\hat{\pi}^{L2}$ to achieve nearly optimal longer term reward for all users in the system with little sacrifice to the short term relevance of responses. In Appendix F.3, we analyze the performance for each of the user groups as well as robustness of MSBL for varying feature noise in user contexts at both levels. We find that MSBL learns the weighted preference of relevance and diversity for each user group and is robust to noisy features.

**Scaling to three levels.** Next, we analyze all three levels and compare short, medium, and long-term expected rewards in Figure 4(b). First, note that the corners of the simplex are red, blue, and brown corresponding to the Level 3 opt. (MSBL), Level 2 opt. (MSBL), and Level 1 opt. policy respectively. Level 3 opt. (MSBL) refers to three nested bandit policies with Level 1 policy obtained via *policy modification* with $a^{L2}$ and Level 2 policy obtained via *feedback modification* with macro intervention $a^{L3}$. The short term optimizing policy performs poorly for both the medium and long term, validating that optimizing engagement leads to sub-optimal performance in the longer term. Level 2 opt. policy (MSBL with 2 levels) performs optimally in the medium term with little sacrifice to the short term but does not perform well for the long term. Level 3 opt. policy (MSBL) achieves the best expected long term reward with little sacrifice to the medium and short rewards. Taking a random intervention at the third level lies strictly inside the Level 3 opt. policy. This experiment validates the scalability of MSBL for more than two levels. In Appendix F.3, we analyze the tradeoff in feedback across all

user groups from all three timescales. We find that MSBL optimizes the long term rewards for each of the user groups.

## 5.3 Recommender System

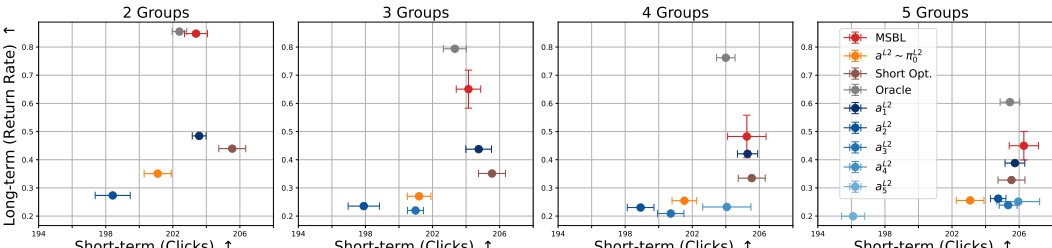

Figure 5: Recommender system: Tradeoff between long term return rate and clicks by **varying groups** using the boost $a^{L2}$ to the relevance scores as **policy modification**. Expected short and long term rewards are reported across 5 random seeds.

In the final setting, we use the KuaiRand dataset [12] to simulate two levels of short and long term feedback. This allows us to evaluate based on real-world user features, and their historical interactions, thereby evaluating for increasing action spaces. We use the simulator developed in [45] and modify it for the contextual bandit setting. We use a training dataset of size 14,265 and test randomly selected 1,771 users. At the lower level, each user logs $T = 5$ interactions, and a transformer model uses item and user embeddings to predict relevance scores $s$. Action $a^{L1}$ represents a top-k selection of items for a context $x^{L1}$, from the policy $\hat{\pi}^{L1} \leftarrow \arg\max_k s$, with the micro reward as average clicks per user.

At the upper level, we simulate a macro intervention $a^{L2}$ as the boost to the scores $s$. This is an instance of *policy modification* since the intervention $a^{L2}$ is applied post optimization to the micro policy. We learn a policy $\hat{\pi}^{L2}$ to select the boost $a^{L2}$ for a given user $x^{L2}$. We simulate user and item groups such that each user group has an unknown preference for a particular item group that is not evident in the micro-level feedback, but only in the macro-level feedback (e.g., less click bait). We simulate the macro reward of return rate as a non-linear function of the long-term preference-weighted fraction of selected items.

**Scaling Level 2 action space.** The purpose of applying a macro intervention $a^{L2}$ is to boost certain item groups for certain user groups. We increase the macro action space by increasing the granularity of user and item groups. As the groups increase, the top-k selected items may not belong to the preferred item groups, making the problem setting harder. Figure 5 shows the tradeoff in the long term user return rate and short term reward of clicks (under the intervention of boost from the macro policy) for top-10 selection. As the number of groups varies $\in \{2, 3, 4, 5\}$, MSBL maintains a high return rate compared to all the baseline policies, with some sacrifice to the short-term clicks. Policies indicated by $a_j^{L2}$ use the same micro policy $\hat{\pi}^{L1}$ as MSBL, but apply the boost only to item group $j$ for all users in the system. The short-term optimization policy $\hat{\pi}^{L1}$ has no macro intervention applied to the scores and maximizes the clicks, but results in low user return rates across all group sizes.

We study the changes in outcomes when the multiscale policies are updated asynchronously after deployment in Appendix F.4. We find that as new data becomes available, updating only the micro policy improves short term outcomes but may deteriorate long term outcomes requiring an update to the macro policy. We also evaluate the robustness of macro learning with varying noise in the micro policy and analyze the tradeoff with increasing Level 1 actions (selection set size). We find that MSBL is robust to perturbations in micro policy, and it outperforms baselines consistently across varying selection sizes.

## 6 Conclusion, Limitations and Future Work

We study the problem of how to train AI systems so that they achieve long-term desirable outcomes. Focusing on the contextual bandit setting, we introduce a MultiScale Policy Learning framework that can use plenty of data at the lower levels as prior information for enabling learning from scarce data at the higher levels. We show how this bridges the disconnect in timescales between short-term

actions and long-term feedback when optimizing for long-term objectives. Furthermore, we show how this framework can be implemented in a practical algorithm, which we found to be effective in optimizing long-term outcomes across a range of domains. However, there are many other ways of instantiating our MultiScale framework with other algorithms, which provide many directions for future work.

**Limitations.** One limitation of this work is the availability of real-world multi-scale datasets in the public domain. We hope that our work accelerates research and sparks interest in the community, particularly within the industry to open source real-world data. Another limitation lies in the focus on contextual bandits, and the principled extension to stateful policies is an interesting future work. Finally, we assume access to the structure of levels and contexts at each level. Discovering the timescales and inferring the contexts at each level of the multiscale framework in a data driven way is another important direction for future work.

# 7   Acknowledgements

This research was supported in part by NSF Awards IIS-2312865 and OAC-2311521. Yuta Saito was supported by the Funai Overseas Scholarship. All content represents the opinion of the authors, which is not necessarily shared or endorsed by their respective employers and/or sponsors. We thank Ben London for helpful feedback on the PAC Bayesian theory, Zhaolin Gao for help with the LLM experiment setup questions, and Taran Singh, Woojeong Kim, and Kiante Brantley for helpful discussions in brainstorming.

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

## A  Broader Impacts

Optimizing for long term outcomes is a desired goal for many interactive AI systems. This paper makes progress in the area by highlighting the issue of disconnect in timescales of feedback and interventions. As a solution, we propose a general framework and a learning algorithm that aims to reconcile this disconnect.

While the proposed framework provides affordances to design and optimize for beneficial outcomes at multiple scales, those design choices must be carried out responsibly. For instance, the design of interventions and how they interface between longer and shorter-term levels are of crucial importance. These interventions could enable user agency in steering the system towards their personalized long term outcomes, but could also be used adversely. Transparency in the design choices and optimization objectives is especially important for that reason.

## B  Additional MSBL Example

Consider a conversational system, consisting of multiple agents, where each agent specializes in a specific type of food cuisine. The platform assists users with various cuisine preferences over a session consisting of multiple queries. Within these preferences, some users prefer more diverse recommendations in a session than others. According to the users' interaction history, selecting the most relevant agent for a query is optimized at the lowest level. The feedback at this level is the relevance of the generated response from a LLM policy, with relevant responses leading to better engagement. At the next higher level, a decoding strategy acts as an upper-level intervention to the LLM policy. The intervention of the decoding strategy can steer the responses towards more or less diverse responses depending on user preferences, leading to a better user return rate on the platform. While optimizing for this return rate seems important, at an even higher level, some users may not prefer to use the platform as heavily as compared to others. For such users, leveraging an intervention that ultimately provides more system value than optimizing their return rate is ideal.

## C  Extended Related Works

Prior works have studied the misalignment between micro and macro objectives to be substantial [5, 28]. Specifically, [5] compares the ranking dcg metric for the micro/short term utility of micro actions (ranking) and the macro violation for the long term constraint violation. Similarly, [28] demonstrate the tradeoff between utility and amortized exposure (collected over a time horizon T) as the macro goal. These works consider optimizing for L1 actions only (rankings). They do not learn L2 interventions such as reward weights [26], ads, message notifications, decoding strategies etc that are available in the system and can directly steer the system towards long term objectives. In contrast, MSPL framework learns these interventions via the macro policy and extends to more than two timescales of competing objectives. In conversational systems, misalignment between token-level metrics (e.g., perplexity) and multiple responses/document-level metrics (e.g., diversity in a paragraph) has been studied [8, 41, 32].

Recent works have explored hierarchy in the action space and arms in bandits [22, 34]. These works do not learn for long-term outcomes but for efficient multi-task learning. [22] proposes a tree-based hierarchical structure, where each tree node represents an action abstraction, grouping similar actions in its child nodes. In contrast, our focus of study is hierarchy in the timescale of feedback, and clear separation of action space at different levels. For long term optimization, [25] proposes a bandit algorithm in the progressive feedback setting, where the long term outcome is increasingly predictable as more short term information is revealed. It models the long term outcome as the delayed outcome of micro level actions. This setting is different than ours as it does not take into account the tradeoff in short and long-term outcomes.

For multi-objective optimization, [6] proposes learning an RL-based policy for the weights of clicks, shares, likes etc. in recommender systems. Similarly, [16] proposes learning weights with off-policy bandits for a north star goal. Different than these works, our framework learns a single policy as a family of policies at the micro level and a separate policy at the macro level. This allows learning with more abundant feedback at the micro level to narrow down the potential actions for the upper level and learn macro policy from relatively sparse data. Our framework also learns contextually and

handles a richer class of reward functions than the linear scalarization approach in [6, 16]. [9] first introduced learning a single model trained with a distribution of losses instead of learning multiple models, each trained with a single loss function. [42] learns a family of models for recommender systems given a distribution of preference vectors. In LLM Alignment literature, [43] introduces learning a family of policies by including preference weighting in the prompt context of LLM policies. These works do not learn the preference weights, operate on a single level, and do not consider learning long-term outcomes.

Related to the adverse effects of over optimization on short term feedback, [30] recently showed that feedback loops within in context learning can cause reward hacking via output and policy refinement. In this paper, we propose policy and feedback modification as two ways to construct priors and mitigate the over-optimization of short term feedback.

# D  MSBL Algorithms

## D.1  MSBL Inference Algorithm

Below, we provide the inference procedure for two levels. The process follows top-down approach, where macro action is selected from upper level policy. This macro action then selects the lower level policy as action $a^{L1} \sim \hat{\pi}^{L1}_{a^{L2}}(.)$

---

**Algorithm 2** MultiScale Inference

---

**foreach** upper level timescale $t2 \in \{1, \ldots\}$ **do**
    Observe upper level context $x^{L2} \sim p(x^{L2})$
    Get upper level action $a^{L2} \sim \hat{\pi}^{L2}(a^{L2}|x^{L2})$
    **foreach** lower level timescale $t1 \in \{1, \ldots T\}$ **do**
        Observe lower level context $x^{L1} \sim p(x^{L1}|x^{L2})$
        Get lower level action $a^{L1} \sim \hat{\pi}^{L1}_{a^{L2}}(a^{L1}|x^{L1})$

---

## D.2  Extending MSBL to $k$ levels

Algorithm 3 presents MSBL extended to multiple levels by calling the PolicyLearning procedure recursively for any two levels. We start with the highest level $k$, and recursively call the next lower level until the base case of k=1 is reached. The Algorithm would then return to the PolicyLearning procedure of the next two upper levels and so on.

---

**Algorithm 3** MultiScale Off-Policy Contextual Bandits (for multiple levels)

---

**Require :** $k \geq 2$
**Procedure** *PolicyLearning*$(\pi_0^{L(k)}, \pi_0^{L(k-1)})$
    **if** $k = 1$ **then**
        **return** ;                  // Base case: when at lowest level
    PolicyLearning$(\pi_0^{L(k-1)}, \pi_0^{L(k-2)})$
    Collect Micro Logged dataset
    $D^{L(k-1)} := \{(x_i^{L(k-1)}, a_i^{L(k-1)}, r_i^{L(k-1)}, p_i^{L(k-1)})\}_{i=1}^{n^{L(k-1)}} \sim \pi_0^{L(k-1)}$
    Learn Micro policies $\hat{\Pi}^{L(k-1)}$(Eq. (6) or (7) using $D^{L1}$)
    Collect Macro Logged dataset
    $D^{L(k)} := \{(x_j^{L(k)}, a_j^{L(k)}, r_j^{L(k)}, p_j^{L(k)})\}_{j=1}^{n^{L(k)}} \sim \pi_0^{L(k)}$
    Learn Macro Policy
    $\hat{\pi}^{L(k)} \leftarrow \arg\max_{\pi^{L(k)}} \hat{V}^{L(k)}(\pi^{L(k)}; D^{L(k)})$ (Eq. (8))
    **return** learnt policies $\hat{\pi}^{L(k)}, \hat{\Pi}^{L(k-1)}$

---

# E Multi-Scale Policy Framework

The PAC Bayes generalization bounds we consider in Eq. (3) are derived for counterfactual risk minimization (e.g., clipped inverse propensity estimator) in [20].

As part of the motivation, we illustrate the following example.

## E.1 Gaussian Model

We consider Gaussian parameterizations, where each policy $\pi(.|.,\theta) \in \Pi$ is defined via a parameter vector $\theta \in \mathbb{R}^d$. We define the target policy distribution as $Q^{L2*} = N(\theta^{L2*}, \Sigma^{L2})$, an uninformed prior distribution $P_0 = N(\theta_0, \Sigma_0)$ for some arbitrary $\theta_0$, and an informed prior $P^{L1} = N(\theta^{L1}, \Sigma^{L1})$ centered at the learned micro policy $\hat{\pi}^{L1}$.

The KL divergence between any two multivariate gaussian $KL(Q^{L2*}||P_0)$ is given by,

$$KL(Q^{L2*}||P_0) = \frac{1}{2}\left[\log\frac{|\Sigma_0|}{|\Sigma^{L2}|} - d + \text{tr}(\Sigma_0^{-1}\Sigma^{L2}) + (\theta_0 - \theta^{L2})^T\Sigma_0^{-1}(\theta_0 - \theta^{L2})\right]$$

where $(\theta_0 - \theta^{L2})^T(\Sigma_0)^{-1}(\theta_0 - \theta^{L2}) = |\theta_0 - \theta^{L2}|_M$ is the squared Mahalanobis distance in the parameter space $\theta$.

Using the above, the gain in the number of samples from using an informed prior $P^{L1}$ instead of an uninformed prior $P_0$ is,

$$n_0 - n^{L2} \propto KL(Q^{L2*}||P_0) - KL(Q^{L2*}||P^{L1}) =$$
$$\text{tr}(\Sigma_0^{-1}\Sigma^{L2}) - \text{tr}((\Sigma^{L1})^{-1}\Sigma^{L2}) + (\theta^{L2} - \theta_0)^T\Sigma_0^{-1}(\theta^{L2} - \theta_0) - (\theta^{L2} - \theta^{L1})^T(\Sigma^{L1})^{-1}(\theta^{L2} - \theta^{L1})$$
$$+ \log\frac{|\Sigma_0|}{|\Sigma^{L1}|} \tag{9}$$

Setting $\Sigma^{L1} := \Sigma_0 = \Sigma_P$ and since the squared Mahalanobis distance is symmetric, we have

$$KL(Q^{L2*}||P_0) - KL(Q^{L2*}||P^{L1}) = (\theta^{L2} - \theta_0)^T\Sigma_P^{-1}(\theta^{L2} - \theta_0) - (\theta^{L2} - \theta^{L1})^T\Sigma_P^{-1}(\theta^{L2} - \theta^{L1})$$

As a result, we can have the sample gain from using the informed prior $P^{L1}$ instead of $P_0$ as

$$n_0 - n^{L2} \propto KL(Q^{L2*}||P_0) - KL(Q^{L2*}||P^{L1}) \in O(|\theta^{L2*} - \theta_0|_M - |\theta^{L2*} - \theta^{L1}|_M) \tag{10}$$

For the special case of isotropic Gaussian distributions, where $\Sigma^{L1} = \sigma^{L1}\mathbb{I} := \sigma_0\mathbb{I} = \sigma_P\mathbb{I}$, Eq. (10) can also be written in terms of L2 distance as,

$$KL(Q^{L2*}||P_0) - KL(Q^{L2*}||P^{L1}) \in O(||\theta^{L2*} - \theta_0||^2 - ||\theta^{L2*} - \theta^{L1}||^2)$$

However, the above is a conservative estimate, since it can be beneficial to pick variances $\sigma^{L1} < \sigma_0$ that increasingly reduce the variance going from uninformed prior $P_0$ to $P^{L1}$.

Next, we simulate a toy example with isotropic Gaussian distributions for $\theta \in \mathbb{R}^{50}$. To calculate $n_0 - n^{L1}$, we use the exact KL divergence in Eq. (9). We start with an uninformed prior variance $\sigma_0 = 200$ and learn all 50 parameters with target variance $\sigma^{L2} = 1.0$. For $n^{L2}$, we start with an informed prior $P^{L1}$ that has 49 parameters learned with $\sigma^{L1} = 1.0$, and we only need to adjust 1 more parameter. The coefficient constant for $\frac{n}{KL(Q||P)}$ is used as $5.0e^3$ for all cases. This provides $\frac{n_0 - n^{L2}}{n_0} = 98\%$ as the reduction in samples needed at the macro level L2. Note that the number of samples used to form the prior $n^{L1}$ is significantly cheaper than $n^{L2}$ since they occur $T$ times as frequently. Empirically, even when taking the additional L1 samples into account, this still provides a $\frac{n_0 - n^{L2} - \frac{n^{L1}}{T}}{n_0} = 88.2\%$ reduction with a $T = 10$ horizon.

# F Experiment Details

For training the bandit policy at a given level, we use Importance Sampling estimator (IPS) [38].

| Level Manager | Algorithm | Estimators | Reward Simulator |
|---|---|---|---|
| Manage levels and their setup | e.g., MSBL algorithm, single stage policy, oracle | e.g., IPS, direct method, doubly robust | Reward functions |

Figure 6: MultiScale simulator modules

For any given level, we use a uniform random policy as the logging policy $\pi_0$, and a softmax policy $\pi(a|x, \theta)$ parameterized by weights $\theta$,

$$\pi(a|x, \theta) = \frac{\exp(\beta\phi(x, a, \theta))}{\sum_{a' \in \mathcal{A}} \exp(\beta\phi(x, a', \theta))} \tag{11}$$

where $\beta > 0$ is the inverse temperature parameter, $\phi(.)$ is a function mapping a given context, action to real value with dimension $d$, parameterized by $\theta$, defined as $\phi(., \theta) : \mathcal{X} \times \mathcal{A} \to \mathbb{R}^d$

IPS is an unbiased estimator and only requires the full support condition, that is $\pi_0(a|x) > 0 \ \forall(x, a) \in \mathcal{X} \times \mathcal{A}$.

For feedback modification, we additionally pass the macro action with the features, so that,

$$\pi^{L1}(a^{L1}|x^{L1}, a^{L2}, \theta) = \frac{\exp(\beta\phi(x^{L1}, a^{L2}, a^{L1}, \theta))}{\sum_{a^{L1'} \in \mathcal{A}^{L1}} \exp(\beta\phi(x^{L1}, a^{L2}, a^{L1'}, \theta))} \tag{12}$$

We sample $a^{L2}$ from a uniform distribution during training. At inference $a^{L2}$ is selected from the learned macro policy.

**Simulator** The simulator code can be found at `https://github.com/RichRast/mspl`. Figure 6 shows the structure of the simulator with four key modules. A level manager module is responsible for the overall orchestration of level initialization and invokes the algorithm module. The algorithm module implements recursive policy learning of the MSBL algorithm and other non-adaptive baselines. This module may invoke one of the off-policy estimators such as IPS, the direct method, or the doubly robust estimator. Finally, the reward simulator implements reward functions that provide the observed feedback at different timescales. Policy learning options for the algorithm, estimators, and the type of reward function are configurable.

**Compute Resources** We use NVIDIA RTX A6000-48GB GPUs for experiments in conversational systems and 24GB NVIDIA GeForce RTX 3090 for experiments in conventional recommender system.

### F.1 A toy example to illustrate comparison with flat RL

We consider a finite horizon setting MDP with the goal of learning $\pi(a|x)$ that optimizes the long term reward $r^{L2} \sim p(r^{L2}|(x_i, a_i), \dots (x_{i+T}, a_{i+T}))$ observed after $T$ timesteps.

We use (tabular) Q learning to learn a policy that optimizes for the sparse reward $r^{L2}$ for the following setup. Users arrive with context $x_t \sim p(x_t)$ and select $k$ out of $n$ items based on relevance vector $r_t(x_t) \in \mathbb{R}^n$. The action $a_t$ can be one out of $k$ choose $n$ combinations. We simulate $n = 10$ items, and 2 contexts $x_t$. The long term reward for each user is observed based on their preference for the item. We use a time horizon of $T = 5$.

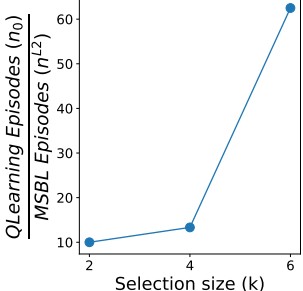

While Q learning is provably optimal, it is computationally expensive to form the Q table by learning from episodes $\{(x_i, a_i), \dots (x_{i+T}, a_{i+T}), r^{L2}\}$, especially when $r^{L2}$ is sparse. In contrast, MSBL leverages the problem structure as follows. Its micro policy is simply argmax on item relevances. Even though this myopic policy doesn't lead to optimal long-term reward, it provides a prior for the macro level, which only needs to learn the macro intervention $a^{L2}$ as the amount of boost to the relevance vector $r_t$. We construct $|\mathcal{A}^{L2}| = 8$, so that $\pi^{L2}$ only needs to select the best $a^{L2}$ for each user context.

Figure 7: Ratio of Q learning vs MSBL episodic samples for a toy setup as the action space increases with $k$.

As a side note, while function approximation and other advancements (actor-critic, etc.) would provide benefit to the RL baseline, they are orthogonal to the key idea of enabling faster learning

via the use of hierarchical priors for sparse long-term feedback. Figure 7 shows the ratio of episodic samples $n_0$ to the MSBL macro policy samples $n^{L2}$ to achieve the same $r^{L2}$ in expectation. We can see that $10 \leq \frac{n_0}{n^{L2}} \leq 62$ as $k$ varies in $\{2, 4, 6\}$. Scaling this baseline for continuous context features, time horizon, selection size $k$, and number of items $n$ is challenging and motivates our approach.

### F.2 Multi-turn conversation

**Experiment Setup.** We use Llama-2-7b-chat [39] as the base model at the lower level. We use huggingface reward models gpt2-large-harmless-reward_model and gpt2-large-helpful-reward_model for harmless and helpful reward models $R_1, R_2$. Following rewards in context learning [43], we train a micro policy as a family of LLM policy $\hat{\Pi}^{L1}$ by including $a^{L2}$ in the context prompt and optimizing $\sum_{i=1}^{2} a_i^{L2} R_i$. The conversation starts from one of the query prompts $x^{L1}$ (a question in the Anthropic dataset), and for each subsequent turn, the trained LLM policy $\hat{\Pi}^{L1}$ responds. This completes one turn. Then the user LLM asks a follow-up question. As a result, each generated response $y_t \sim \pi_{a^{L2}}^{L1}(.|x_{<t}^{L1}, y_{<t})$ consists of the conversation upto that turn. For the user LLM, we use another Llama-2-7b-chat model. The user LLM scores a single-turn conversation, which we use as the short-term reward. This is an instance of feedback modification that steers the LLM policy using modeled feedback (where feedback is defined by a model). This differs from the observed feedback (i.e, user LLM evaluations, which are not modeled and only observed for an action that is taken). At the upper level, we learn interventions $a^{L2} \in [0.8(\text{harml-}), 0.2(\text{helpf-})], [0.2(\text{harml-}), 0.8(\text{helpf-})]$. $a_1^{L2}$ refers to the same level 1 policy as other baselines but using the fixed level 2 intervention of preference weights in [0.8(harml-), 0.2(helpf-)], while $a_2^{L2}$ refers to using the fixed level 2 intervention of preference weights in [0.2(harml-), 0.8(helpf-)].

We use the upper level reward simulator as Llama-3-70b API [13], which scores user satisfaction for the given user persona after a full 5-turn conversation as a binary score in $\{0, 1\}$. We transform the user persona text with MPNet model [36] and project down with PCA to a 10-dimensional embedding. This forms the upper-level context $x^{L2}$ for bandit policy $\pi^{L2}$.

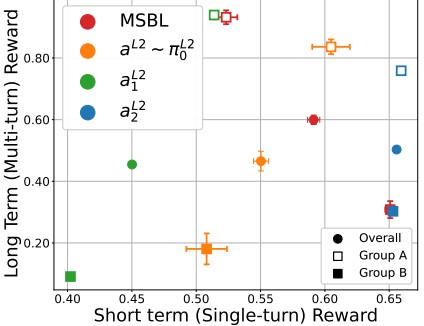

**Group-wise Results** Figure 8 shows the long term multi turn reward vs per-conversation short term reward for each of the user personas - Child and Expert. We note that MSBL achieves the best multi-turn reward for each user persona while the other policies are optimal only for one type of user group.

Figure 8: Long-term (multi-turn) vs short-term (single-turn) rewards for all users (overall) and for each user group.

**Training Details.** For $\pi^{L2}(.|x^{L2})$, we use a 5 layer neural network with hidden dimension 256. We train using AdamW optimizer, with a batch size of 256, learning rate 1e-3, weight decay 1e-1 for 1000 epochs. We use the inverse temperature parameter $\beta = 0.8$ in Eq. (11).

**Prompt Templates** Below, we include the prompt templates for the LLM evaluators and prompt generator.

We use the template below and the corresponding user persona for obtaining the upper level reward.

```
### User Persona Prompts
    child: You are a child user who wants to avoid harmful ,
        illegal , or unethical content .,
    expert: You are an expert user who values accurate and
        relevant content .

### Llama -3 -70b Evaluation Template
system_prompt = You are an expert conversation evaluator . Given
    a user personality and conversation , output only a score 0
    or 1 indicating user satisfaction . Repetition of responses
    in the conversation should not have any impact on your
    evaluation .
User Personality : {user persona}
```

```
output format:
score: [0.0 or 1.0]
Here are some examples of how you might respond:
{Examples}

template = {role: system,
content: {system prompt},
role: user,
content: Conversation to evaluate:{conversation}
Evaluate user satisfaction. Please provide a score and
    explanation in the required format.}
```

User Llama-2-7b Evaluator and Prompt Generator

For the next prompt generation, we use the below template with the history of previous turn conversation.

```
### User Llama-2-7b Evaluator and Prompt Generator
system prompt = You are a user having a conversation with an AI
    assistant.
{user persona}
Given the conversation history, generate only the next user
    message or question (one or two sentences) to continue the
    conversation.
output format:
User: [next user message]

template = { role: system,
content: {system prompt},
role: user,
content: Conversation history:{conversation}
Respond with just your next message in the exact format.
}
```

For evaluating a single turn, we use the below template. This score serves as the simulated lower level reward.

```
system prompt = You are evaluating an AI assistant's response
    as a user.
{user persona}.
You are provided with a user's prompt and an assistant's
    response. Rate the usefulness and quality of the response
    from 0.0 to 1.0.
You must use this exact format for your response:
score: [between 0.0 and 1.0]

template = { role: system,
content: {system prompt},
role: user,
content: User: {prompt}
Assistant: {response}
Provide score in the required format.
}
```

## F.3    Conversational recommender system

**Experiment Setup** At each of the three levels, we simulate 5-dimensional contexts sampled from a normal distribution $\mathcal{N}(\mu_g^l, \sigma_f)$ belonging to two user groups. The context $x^{L1}$ represents users' demographics and cuisine preferences. At Level 1, there are 10 bandit actions $a^{L1}$, corresponding

to a cuisine:{"Ethiopian", "Mexican", "French", "Japanese", "Spicy Indian", "Thai", "Carribean", "Peruvian", "Russian", "Italian", }. We use Llama-2-7b-chat [39] as the LLM policy $\pi^{LLM}$ that acts as a personalized agent, generating cuisine suggestions $y$ to users at L1 timescale $t1 = \{1, \ldots 10\}$. Each query consists of a system prompt specifying the agent's expertise and a user query $q$. We learn a bandit policy $\pi^{L1}(a^{L1}|x^{L1})$ that selects the particular LLM agent $a^{L1}$. Next, we generate response $y_t \sim \pi^{LLM}(.|a^{L1}, q, y_{t-1})$ from the LLM policy, given the agent selection $a^{L1}$ and append the previous timestep response $y_{t-1}$ to the current query. The L1 reward is simulated as inverse perplexity conditional on the optimal prompt, representing the engagement (relevance) metric, as follows

$$r_t^{L1} = \text{perplexity}(y_t|a^{L1*})^{-1} = \exp\left(\frac{1}{\text{\# tokens in response } y_t} \sum_i^{\text{\# tokens in } y_t} \log \text{prob}(y_{t,i}|y_{t,0:i-1}, a^{L1*})\right)$$

where $a^{L1} = \{\text{cuisine}\}$ is selected according to $\pi^{L1}$, and the optimal action $a^{L1*}$ corresponds to the optimal prompt according to user preference.

Intuitively, the inverse perplexity metric means that the response $y_t$ generated using the bandit action $a^{L1}$ should give the highest perplexity if the optimal action $a^{L1*}$ is chosen. For example, if the user's preference was french, but the bandit policy $\pi^{L1}$ selected an action corresponding to mexican cuisine, then the response generated $y$ consisting of mexican cuisine would have a low perplexity given the prompt of french cuisine (i.e, the optimal action $a^{L1*}$)

The query $q$ at every timestep $t \geq 1$ is as follows,

```
You are my personal chef experienced in {cuisine} Cuisine. Your
    responses should be professional and concise (100 words or
    less). Previously, you suggested: {previous generated
    response}.
What should I eat today?
```

At the second level, we simulate two user groups with context $x^{L2}$ representing preferences for engagement (relevance) and diversity. We use the notion of $n$-gram repeats over multiple responses ($t1 \in \{1, \ldots 10\}$) for computing diversity, where we use $n = 3$. In particular, we define diversity as normalized 3-gram repeats, where the 3-gram repeats are computed over $T = 10$ responses generated at Level 1 for each user. We simulate the reward

$$r^{L2} = \sigma\left(\beta_u\left(\frac{\sum_{t1=1}^{10} r_{t1}^{L1}}{10}\right) + (1 - \beta_u)(\text{diversity})\right)$$

, where $\sigma(x) = a(x - b)$ is scaled sigmoid function with scaling factors $a = 60, b = 0.6$. $\beta_u$ is the unknown user's relevance-diversity trade-off parameter, and we use $\beta_u = \{0.9, 0.1\}$ for groups that prefer relevance and diversity respectively. The second level policy $\pi^{L2}$ selects an intervention of decoding temperature $a^{L2} \in \{0.0, 0.2, 0.4, 0.6, 0.8, 1.0\}$ as a *policy modification* to the Level 1 policy, with $a_1^{L2} \ldots a_6^{L2}$ referring to each of the L2 interventions.

Some sample responses (decoding temperatures are logged after generation) are,

```
"Hola! How about our delicious carne asada burritos with sauted
    onions and bell peppers, topped with melted cheese and
    served with a side of sliced avocado and a drizzle of
    homemade salsa? decoding_temp_1.10",
"For lunch today, I recommend our 'Chili con Carne Burrito.' It
    's a satisfying blend of slow-cooked beef, beans, and
    vegetables wrapped in a large flour tortilla, topped with
    creamy sour cream and fresh cilantro. It's a hearty and
    flavorful dish that will warm you up on a chilly day! "
    decoding_temp_0.87",
"Great! Based on your previous preferences, I recommend trying
    our 'Shiro Wot' - a creamy stew made with chickpeas, onions
    , garlic, and a blend of spices. It's a popular vegetarian
    dish that's sure to satisfy your taste buds. Pair it with a
     side of 'Kik Alicha' - sauted vegetables in a mild sauce,
    and enjoy! decoding_temp_0.21",
```

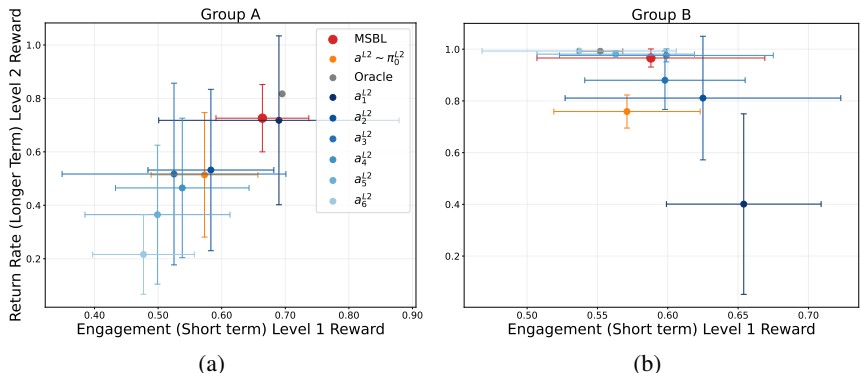

(a)             (b)

Figure 9: Conversational recommender system: Tradeoff between longer-term Level 2 and short-term Level 1 rewards using decoding temperature $a^{L2} \in \{0.0, 0.2, 0.4, 0.6, 0.8, 1.0\}$ as **policy modification** across 5 random seeds. (a) for all users (b) for users belonging to group A that prefer relevance (c) for users belonging to group B that prefer diversity among multiple responses

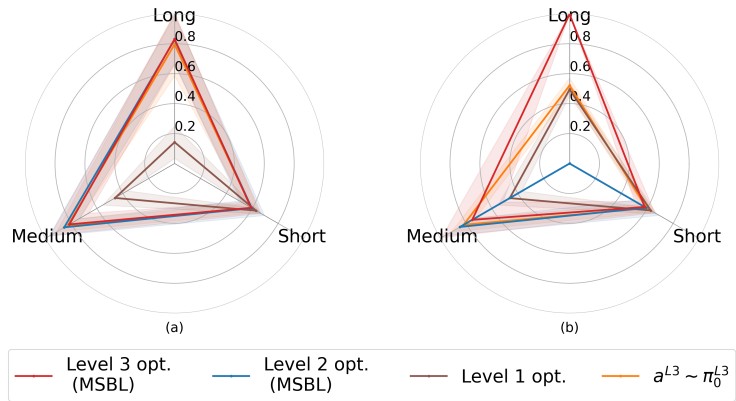

Figure 10: Tradeoff in different timescale rewards for the two user groups at Level 3. (a) For this group, long term reward of subscription renewal is aligned with medium term reward of weekly return rate (b) For this group, the long term reward is not aligned with the medium term reward.

At the third level, we simulate users with context $x^{L3}$. We use *feedback modification* for Level 2 training to get a family of policies $\hat{\Pi}^{L2} := \{\hat{\pi}^{L2}_{a^{L3}} : a^{L3} \in \{[0,1], [1,0]\}\}$. The parameterized feedback is given by $\sum_{i=1}^{2} a_i^{L3} R_i$, for two reward models $R_1, R_2$, such that $R_1 = r^{L2}$, and $R_2 = \min(\tau, r^{L2})$. We use $\tau = 0.8$ as a user group-specific threshold. This simulates the scenario where for one user group, optimizing the weekly return rate is beneficial, while for the other user group, optimizing only upto a threshold $\tau$ is beneficial. We simulate $r^{L3}$ across 2 timescales of level 2 as follows,

$$r^{L3} = \gamma_u \frac{\sum_{t2} r_{t2}^{L2}}{2} + (1 - \gamma_u) \, \mathbf{1}(\text{activity preference})$$

where $\gamma_u \in \{1, 0\}$ is the trade-off parameter.

**Training Details** For bandit policy $\pi$ at each level, we use a 3 layer neural network with hidden dimension 256. We train using AdamW optimizer, with a batch size of 256, learning rate 1e-4, for 4000 epochs.

**MultiScale contexts.** In this experiment, we evaluate across user groups from all three timescales. At each level, we simulate contexts belonging to two user groups.

Figure 9 shows the tradeoff in longer-term L2 reward and shorter term L1 reward for the two user groups individually. The context $x^{L2}$ at Level 2 belongs to these two groups. Similarly, Figure 10 shows the tradeoff in different timescale rewards for the two groups that $x^{L3}$ belongs to at Level 3.

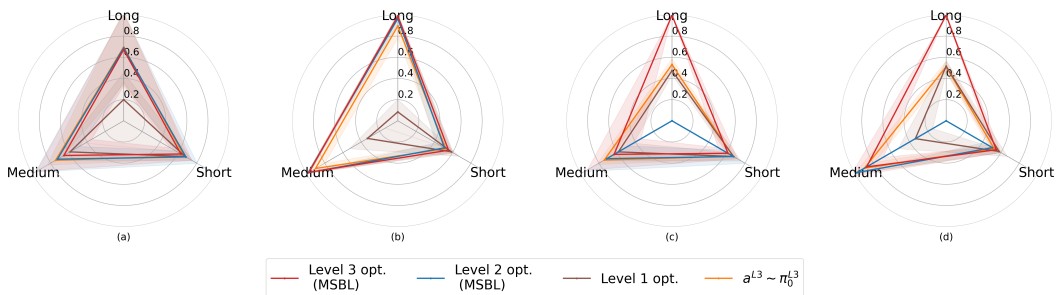

Figure 11: MultiScale contexts ($x^{L2} \mid x^{L3}$) and rewards for 4 groups in (a), (b), (c), and (d) across all three timescales.

| Noise | | | | | Policy | | | |
|---|---|---|---|---|---|---|---|---|
| | MSBL | $a^{L2} \sim \pi_0^{L2}$ | $a_1^{L2}$ | $a_2^{L2}$ | $a_3^{L2}$ | $a_4^{L2}$ | $a_5^{L2}$ | $a_6^{L2}$ |
| 0.05 | **0.82±0.13** | 0.66±0.10 | 0.46±0.13 | 0.68±0.16 | 0.76±0.11 | 0.77±0.10 | 0.72±0.08 | 0.64±0.06 |
| 0.10 | **0.86±0.08** | 0.66±0.09 | 0.48±0.10 | 0.61±0.13 | 0.77±0.12 | 0.74±0.11 | 0.75±0.11 | 0.62±0.06 |
| 0.20 | **0.76±0.10** | 0.57±0.08 | 0.43±0.18 | 0.59±0.10 | 0.60±0.08 | 0.61±0.11 | 0.58±0.08 | 0.55±0.06 |
| 0.30 | **0.66±0.12** | 0.54±0.06 | 0.34±0.17 | 0.49±0.12 | 0.49±0.12 | 0.55±0.03 | 0.53±0.04 | 0.53±0.02 |
| 0.40 | **0.58±0.06** | 0.53±0.02 | 0.37±0.09 | 0.45±0.12 | 0.51±0.02 | 0.51±0.04 | 0.52±0.01 | 0.51±0.01 |
| 0.50 | **0.62±0.08** | 0.47±0.03 | 0.33±0.09 | 0.50±0.05 | 0.51±0.01 | 0.52±0.02 | 0.52±0.01 | 0.51±0.01 |

Table 1: Level 2 Reward with **varying noise in features**

In both cases, we find that MSBL policy achieves near-optimal long term rewards for each of the groups.

A Level 2 context $x^{L2}$ is drawn given a Level 3 context $x^{L3}$. For example, given that a user does not want to be on the app every week (Level 3 preference), the user likes more diverse responses (Level 2 preference). This results in 4 user groups across L2 and L3. Figure 11 shows the tradeoff in expected rewards across these 4 groups. While there are user groups for whom a policy that is optimal in the medium term is also near-optimal in the long term (groups 1 and 2 with overlapping red and blue lines), Level 3 opt. (MSBL) pareto dominates each of these policies for all the 4 groups for the long term reward.

**Robustness to noisy features.** We also evaluate robustness of MSBL with varying noise in the user features with $\sigma_f \in \{0.05, \mathbf{0.1}, 0.2, 0.3, 0.4, 0.5\}$ with default value in bold. Table 1 shows that while the expected Level 2 rewards decrease with increasing noise in features, MSBL maintains the advantage of learning the interventions over other baselines across noise variations.

### F.4 Recommender System

In the final setting, we use the KuaiRand dataset [12] to simulate two levels of short and long term feedback. This allows us to evaluate based on real-world item, user features, and their historical interactions, thereby evaluating for increasing action spaces. We use the simulator developed in [45] and modify it for the contextual bandit setting. We use a training dataset of size 14,265 and test randomly selected 1,771 users. There are 5659 items and their descriptions/features in the dataset. At the lower level, each user logs $T = 5$ interactions, and a transformer model uses item and user embeddings to predict relevance scores $s$. Action $a^{L1}$ represents a top-k selection of items for a context $x^{L1}$, from the policy $\hat{\pi}^{L1} \leftarrow \arg\max_k s$. Then, we simulate clicks according to a Bernoulli distribution of the relevance scores of selected items and micro-level reward as average clicks per user.

We form item groups based on the upload type of the video and user groups based on the users' activity level feature. At the upper level, each user group has an unknown preference $p_{u,i}$ for a particular item group. We use $p_{u,i} = 0.9$ for the preferred user and item group pair. The user retention probability $p_r$ is simulated as this preference-weighted fraction of items selected, given by $p_r = \text{softmax}\left(\log\left(\sum_i p_{u,i} \frac{n_i}{\sum_i n_i}\right)\right)$, where $n_i$ is the number of items selected from group $i$ over

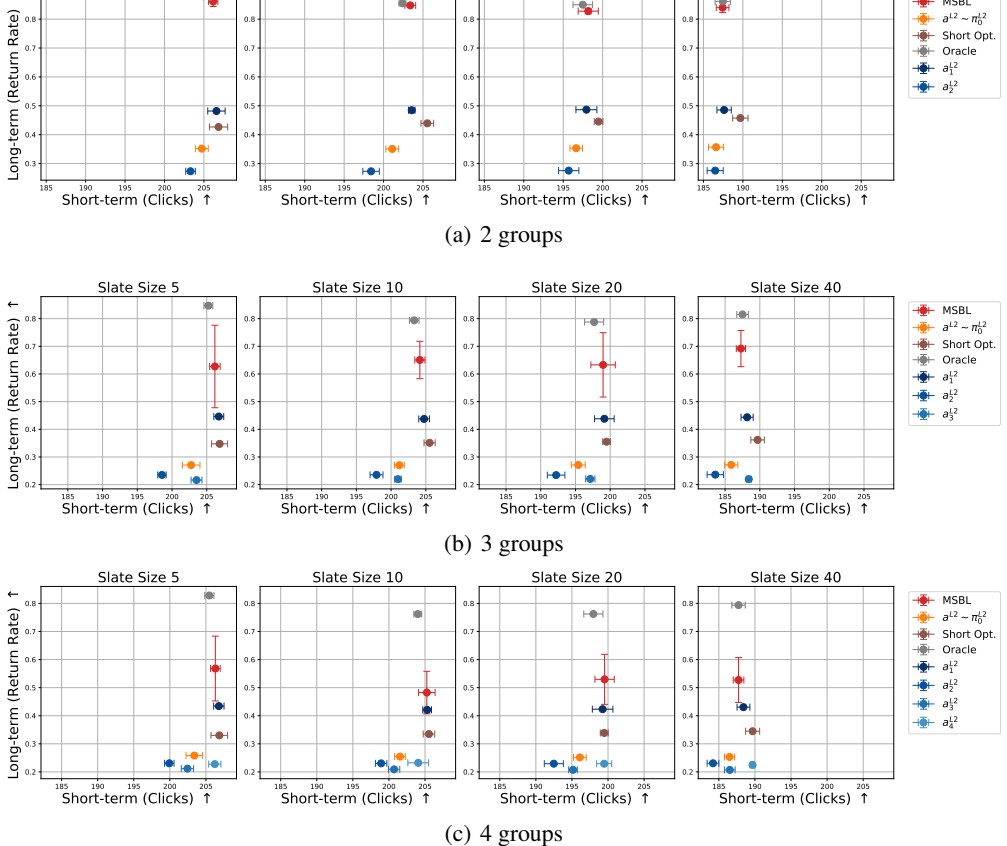

Figure 12: Recommender system: Tradeoff between long term return rate and clicks by **varying slate size** using the boost $a^{L2}$ to the relevance scores as **policy modification** across 5 random seeds when the **number of groups vary** in (a) 2 groups (b) 3 groups and (c) 4 groups.

$t1 = \{1, \ldots, 5\}$ timesteps of the lower level. The reward $r^{L2}$, representing the return rate is simulated as the inverse of the return day to the app.

For training the lower level policy, which is a transformer model that uses input as the item and user embedding and predicts relevance scores, we use the user features and pre-processing following [45].

For the upper level bandit policy, we use 5 user features, namely 'uf_user_active_degree','uf_is_live_streamer','uf_is_video_author', 'uf_follow_user_num_range', 'uf_fans_user_num_range', with their definitions provided in [12].

**Training Details** For bandit policy $\pi^{L2}$ at upper level, we use an embedding module that embeds the raw user features into 16-dimensional vectors and a 4 layer neural network with hidden dimension 128. We train using AdamW optimizer, with a batch size of 128, learning rate 1e-4, for 4000 epochs.

**Increasing Level 1 and Level 2 action space.** Figure 12 shows the tradeoff with **varying groups** in $\{2, 3, 4\}$ and with varying Level 1 actions as the slate size top-$k$ varies for $k \in \{5, 10, 20, 40\}$. We observe that the return rate for MSBL remains high across all sizes and for all groups. Since we report average clicks per user, there is a decrease in the short term (clicks) with increasing slate size. This experiment demonstrates that even with large micro action space $\mathcal{A}^{L1}$, MSBL can effectively leverage the interventions to drive the system toward the long-term reward.

**Updating policies after deployment.** To study the changes in outcomes when the multiscale policies are updated asynchronously, we conduct the following experiment. We use 60% data to learn micro and macro policies and refer to them as the deployed policy. We include an additional 20% data of users and their corresponding interaction history. We update only the micro policy with the increased logged micro data, and leave the macro policy unchanged. The top row of Table 2 shows the % change relative to the deployed policy. The change in return rate under the updated micro policy

|  | 2 Groups | | 3 Groups | | 4 Groups | | 5 Groups | |
|---|---|---|---|---|---|---|---|---|
| % Improv. w. updating | LT | ST | LT | ST | LT | ST | LT | ST |
| only micro policy | -1.18% | **2.47%** | -0.70% | **2.61%** | -0.07% | **3.04%** | 4.88% | **2.85%** |
| micro & macro policy | **0.67%** | 2.34% | **4.90%** | 2.37% | **2.12%** | 2.86% | **5.66%** | 2.66% |

Table 2: Improvement (relative to deployed policy) after policy updates. Short-term (ST) clicks highlighted when only micro policy is updated. Long-term (LT) return rate highlighted when macro policy is also updated.

|  | 2 Groups | | 3 Groups | | 4 Groups | |
|---|---|---|---|---|---|---|
|  | LT | ST | LT | ST | LT | ST |
| w. new macro data | 0.79 ±0.05 | 202.0 ±0.67 | 0.58 ±0.09 | 203.3±0.53 | 0.50±0.07 | 203.8±0.81 |
| w. updating only macro policy | 0.84 ±0.01 | 201.9±0.70 | 0.61±0.08 | 203.1±0.57 | 0.54±0.07 | 203.4±0.84 |
| % Improv. | 5.89% | -0.02% | 6.26% | -0.09% | 8.70% | -0.18% |

Table 3: % Improvement after only macro policy update.

and the clicks under the intervention of the deployed macro policy are reported. While the clicks consistently improve, the return rate can deteriorate, as seen for two groups. Next, we update the macro policy with the logged data collected under the updated micro policy. We compare the % change relative to the deployed policy and find that the long term return rate improves, as seen in the second row of Table 2. In conclusion, while updating the micro policy improves the short term outcome as expected, it can also affect the long term outcome, requiring an update to the macro policy.

Next,we evaluate the long and short-term outcomes when new macro data is available, which we simulate by changing the user preferences at the macro level in the environment. In the first row of Table 3 there is no change to the macro policy while in the second row, the macro policy is updated using the new logged macro data and the micro policy is kept unchanged. We find that the long-term improves without much change to the short-term clicks under the intervention of boosts from this updated macro policy.

**Robustness to errors in the micro policy.** To evaluate the robustness with varying errors in the micro policy, we simulate a perturbed policy $\tilde{\pi}_{a^{L2}}^{L1} := \arg\max_k \tilde{s}$, where noisy relevance scores $\tilde{s}$ are sampled from $\mathcal{N}(s, \sigma_s^2)$, with $\sigma_s \in \{0.0, 0.2, 0.4, 0.6, 0.8, 1.0, 2.0\}$. Figure 13 shows long term reward as this noise varies for top-10 selection with 2 groups. We find that learning the macro policy is robust for fairly large noise $\sigma_s$ and only begins to deteriorate for high values of $\sigma_s$. This experiment also demonstrates the advantage of learning micro policy to the long term return rate metric. In real-world deployment, the micro policy may be updated periodically, so this robustness to errors makes the nested bandit learning practically appealing.

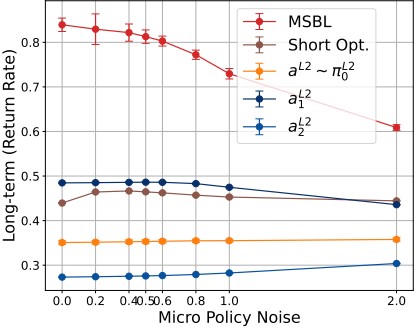

Figure 13: Long term return rate with **varying noise** $\sigma_s$ in micro policy varies in $\{0.0, 0.2, 0.4, 0.6, 0.8, 1.0, 2.0\}$.

