# OpenReview forum: "MultiScale Contextual Bandits for Long Term Objectives"
_NeurIPS.cc/2025/Conference — NeurIPS 2025 poster_

### Official Review · Reviewer_enko · 2025-07-03

**Clarity:** 3
**Significance:** 3
**Originality:** 4
**Rating:** 5
**Confidence:** 3

**Summary:**

The paper considers the problem of mismatched proxy rewards and a natural tradeoff that occurs in most practical settings -- the sparsity of high fidelity long term reward and a more dense proxy reward that might be easier to optimize against. The authors propose viewing this as optimizing a multi-scale policy, where the lower level effectively provides a data driven prior that the sparse higher level policy can tractable guide using the limited feedback available. The authors motivate this suggestion using a simple intuition from PAC-Bayes. The paper proposes two concrete mechanisms for this hierarchical linkage:
Policy Modification: A base policy is first learned to optimize the short-term reward. The higher-level policy's actions then consist of applying modifications to this base policy, such as boosting the scores of certain items in a recommendation list.
Feedback Modification: The higher-level action specifies weights on the objective of the lower-level policy.

The paper considers several experimental analyses and instantiations of the proposed high level idea, where they find that this does lead to plausible improvements over natural baselines. A limitation is the synthetic nature of the tasks studied, which the authors acknowledge clearly.  The paper's primary contribution is the "MultiScale Policy Learning"  framework itself—a new conceptual model for tackling long-term objectives. The algorithm presented, MSBL, uses standard off-policy contextual bandit learning at each level of the hierarchy. The technical innovation lies not in creating a new point-wise learning algorithm, but in the architecture that connects these standard components. The novelty is in the two proposed linkage mechanisms.

**Questions:**

- The definition of MSP appears like a product of two distributions when presented initially in Page 4 (section 4), but at this point, it is a bit unclear whether what is meant is conditional factorization. Could you clarify?

- In Figure 3, I understand the superscript L2 indicating the level, but what are the action subscripts 1 and 2?

**Ethical Concerns:**

["NO or VERY MINOR ethics concerns only"]

**Final Justification:**

Conceptually novel work with reasonable empirical evaluation considering the limits of the available open source datasets. Authors have also addressed most concerns during the rebuttal, so I raise my score to an accept.

**Limitations:**

Yes.

**Quality:**

3

**Strengths And Weaknesses:**

Strengths
- The problem being addressed is ubiquitous in practical applications, and so quite important.
- The proposed hierarchical separation is quite general and powerful in capturing a wide range of situations.
- The empirical validation is extensive. The paper evaluates three distinct and relevant tasks: a multi-turn conversation task, a simulated conversational recommender system, and a recommender system built on a video streaming benchmark dataset.

Weaknesses
- It is unclear how the proposed approach performs against strongly tuned baselines in real world systems, which reasonably optimize various levels of the stack independently at any each point in time. The paper's baselines, such as the "Level 1 opt" policy (optimizing only short-term reward) or a policy with random macro-level actions, appear designed to clearly demonstrate the benefit of the hierarchical learning structure. However, a sophisticated real-world system might employ a more complex baseline, perhaps by training separate models for different objectives and combining their outputs with a hand-tuned business logic layer.  While the authors propose that their framework can be seen as learning the macro-level interventions that might otherwise be fixed or hand-tuned, a direct comparison to a strong, practical baseline is missing (for good reasons, since the study is not performed on a production system).

- Technical novelty limited (even if the conceptual contribution is significant), with standard off the shelf off policy learning being the workhorse underneath.

---

> ### Author Rebuttal · Authors · 2025-07-31
>
> We thank the reviewer for their time and effort for this valuable service.
>
> **We are glad that all four reviewers unanimously acknowledge the novelty of the method with high scores along the axis of Originality** and the reviewer noted “The proposed hierarchical separation is quite general and powerful in capturing a wide range of situations.” Reviewer mEmM also noted “The hierarchical treatment of feedback across multiple timescales is novel, providing a compelling approach to reconciling short-term actions with long-term outcomes.”
>
> The concerns raised are primarily clarifications. We address each of them below.
>
> > empirical evaluation and tasks
>
> We are glad that the reviewer noted “The empirical validation is extensive. The paper evaluates three distinct and relevant tasks: a multi-turn conversation task, a simulated conversational recommender system, and a recommender system built on a video streaming benchmark dataset.”
>
>
> - Regarding the synethtic nature of the tasks studied, we emphasize these are semi-synthetic experiments and the best available option in the absence of open source real world datasets for this problem. Below we detail how each of the tasks are formed from either real world data where possible (level 1) or simulated according to existing literature (for medium and long term feedback).
>     1. Our first experiment on multi-turn conversations is based on Anthropic’s helpful-harmless dataset [4] and 7b model similar to those in [1]. <!-- The difference from [1] is that [1] does not address or take into account the long term feedback.  -->
> To obtain long term feedback on this task, we set up a user-agent multi turn conversation where the user is instantiated with a LLama-70B. This multi-turn setup is used in [3]. We let the conversation continue for up to 5-turns and obtain the long term feedback from LLama-70B with LLM as judge [6].
>
>     2. The second task, where we built a simulator to demonstrate three levels for a conversational recommender system, is again based on the 7b model as in [1]. We model medium and long term feedback, similar to the long term constraints formulation in [2], where the long term constraints are only observed after T steps and the constraint satisfaction depends on all the actions taken during that time period. In [2], the authors used Kauirand dataset [5] and formed long term constraints as a function of exposure targets. In our work, we similarly form long term feedback based on user preferences. The difference from [2] is that while [2] assumes the long term intervention as given, we learn it with L2 policy.
>
>   3. The third task based on a real world video streaming app uses the dataset of engagements collected from real user interactions [5]. This experiment uses real user features and their interactions with a catalog of ~5K items for level 1. We formulate the long term feedback similar to that in [2].
>
>         [1] Rewards-in-Context: Multi-objective Alignment of Foundation Models with Dynamic Preference Adjustment (Yang et al'24)
>
>         [2] Ranking with Long term constraints (Brantley et al'24)
>
>         [3] Regressing the Relative Future: Efficient Policy Optimization for Multi-turn RLHF (Gao et al'25)
>
>         [4] Anthropic’s helpful-harmless dataset
>
>         [5] KuaiRec dataset for recommender systems (Gao et al'22)
>
>         [6] LLM as judge
>
>
>
> - While we agree that deployment in a production system would be an interesting next step, our key contribution is the conceptualization of the new multi-scale formulation, which the reviewers acknowledge as interesting and important - and for which we provide both a PAC-Bayesian motivation and extensive empirical evaluation of semi-synthetic data. Whether this approach performs well on a particular industrial application cannot be a requirement for publication, and most NeurIPS papers do not provide such experiments.
>
> > strongly tuned baselines in real world systems
>
>
> - Heavily engineered production systems over many years can get excellent performance through manual tuning and complex sets of business rules for effective long term behavior. We kindly emphasize that is not the point of our research. The goal of our research is to provide a principled and general method for learning systems with effective long-term behavior for a variety of domains. To validate this, we provide both a PAC-Bayesian motivation and extensive empirical evaluation on semi-synthetic baselines.
>
>
>
> - In particular, we formed the baselines similar to other accepted papers in the community [1,2]. In [1] the authors provide baselines as the base LLM policy with and without their proposed module of predicting personalized temperature. [1] does not address or model the problem of long term objectives. In [2], the baselines consist of myopic controllers (short term optimizing baseline) and other proposed controllers that optimize for different formulations of long term constraints similar to our baselines for different types of macro interventions.
>
>     [1] To Cool or not to Cool? Temperature Network Meets Large Foundation Models via DRO (Qiu et al'24)
>
>     [2] Ranking with Long term constraints (Brantley et al'24)
>
> > Conceptual vs technical novelty
>
> We thank the reviewer for acknowledging the conceptual novelty of our proposed framework that is general and powerful for optimizing long term outcomes. The reviewer also clearly described our contributions of proposing two ways of building a family of policies at micro level (policy and feedback modifications).
>
> As we noted in the conclusion, there are many other ways of instantiating the general MSPL framework with other algorithms (and potentially new estimators) as an interesting direction for future work.
>
> > Factorization of policies
>
> Thanks for raising this important point and letting us clarify. In section 4, we introduce multiscale policies by factoring a monolithic policy space $\Pi$  as $\Pi^{L1} \cdot  \Pi^{L2}$. By this, we meant that we can decompose the policy space $\Pi$ into decoupled and independent learning at each level.
>
> As shown in Figure 2(b), learning a family of policy from the entire policy space at Level 1 ($\Pi^{L1}$) consists of finding a set $\hat{\Pi}^{L1}= \{\hat{\pi}^{L1}_{a^{L2}}; a^{L2} \in \mathcal{A}^{L2}\}$. The entire policy space $\Pi^{L1}$ can be viewed as consisting of different sets of $\hat{\Pi}^{L1}$.
> At level 2, we find a policy $\pi^{L2} \in \Pi^{L2}$ as shown in Figure 2 ( c ).
> The factorization refers to learning a family of policy at L1 decoupled from learning $\pi^{L2}$ at level 2.
>
> During inference as shown in Figure 2(a) the selection is conditional, where $a^{L2}$ is first selected from the learned $\pi^{L2}(a^{L2}|x^{L2})$ and the lower level action is selected according to $\pi_{a^{L2}}^{L1} (a^{L1}|x^{L2})$
> This does indeed make the overall action selection a conditional product of policies $\pi^{L2}$ and $\pi^{L1}_{a^{L2}}$.
>
> We will add this clarification in section 4 of the main text.
>
>
> > Action subscripts in Figure 3
>
> The action subscripts in Figure 3 stand for static L2 interventions, where $a^{L2} \in$  [0.8(harml-), 0.2(helpf-)], [0.2(harml-), 0.8(helpf-)].
> $a^{L2}_1$ referes to the same L1 policy as other baselines but using the fixed L2 intervention of preference weights in [0.8(harml-), 0.2(helpf-)] , while $a^{L2}_2$ refers to using the fixed L2 intervention of preference weights in [0.2(harml-), 0.8(helpf-)]. We describe this in Appendix F.2 but will clarify it in the main text.

---

> > ### Author Response · Authors · 2025-08-05
> >
> > Could the reviewer let us know whether our response has clarified the raised concerns, and whether this changes the reviewers evaluation of the paper.

---

> > ### Comment · Reviewer_enko · 2025-08-05
> > **reply**
> >
> > Thank you for the clarifications to the notation and other technical details. Many of my concerns have been more or less addressed and I will update my score accordingly.

---

> > > ### Author Response · Authors · 2025-08-07
> > >
> > > Thats great and thanks again for taking the time and effort to engage and provide feedback.

---

### Official Review · Reviewer_z6c1 · 2025-07-03

**Clarity:** 3
**Significance:** 2
**Originality:** 3
**Rating:** 4
**Confidence:** 4

**Summary:**

The authors tackle a common dilemma in interactive systems such as recommender platforms and chat-bots: logs are rich in short-term feedback (clicks, single-turn ratings) but the retention, satisfaction, subscription renewal unfolds on a much slower time-scale. This paper introduce MultiScale Policy Learning, a hierarchical framework that: (1)learns micro-level policies on the plentiful fast feedback; (2)treats each macro action as selecting or modifying one of those micro policies, and (3) learns a macro-level policy—and potentially further levels—using the sparse long-term signals.

**Questions:**

See Weakness.

**Ethical Concerns:**

["NO or VERY MINOR ethics concerns only"]

**Final Justification:**

After rebuttal, the authors have addressed the majority of my concerns, I have increased the score from 3 to 4.

**Limitations:**

Yes.

**Paper Formatting Concerns:**

None.

**Quality:**

2

**Strengths And Weaknesses:**

Advantages


(1) By re-using micro-level knowledge as a Bayesian prior, MSBL needs dramatically fewer long-term samples to reach reliable macro policies—crucial when retention or renewal labels are rare.
(2) The algorithm is a lightweight wrapper around standard off-policy bandits; it supports multiple number of levels and both “policy-modification” and “feedback-modification” ways of generating the micro-policy family. This makes it easy to plug into existing ranking, LLM decoding or recommendation stacks.
(3) Across chat, simulated and real-world recommender tasks, MSBL consistently lifts long-horizon objectives while keeping click/engagement drops small and showing resilience to noisy features or larger action spaces.



Weaknesses:

(1) Lack of empirical support for the core motivation. The central motivation of the paper—that there exists a substantial disconnect between the timescale of user interactions (e.g., click or watch-time) and long-term objectives (e.g., subscription renewal)—is intuitively plausible but insufficiently substantiated. The manuscript lacks quantitative evidence demonstrating how frequent or significant this misalignment is in practice? In many real-world applications, short-term metrics often correlate well with long-term outcomes, which the paper itself implicitly assumes when using L1 feedback to construct priors for L2 learning.

(2) The proposed multi-scale learning framework introduces an explicit hierarchy to separate short-term and long-term objectives. However, in many bandit settings, long-term feedback can be modeled as delayed rewards and incorporated directly into the L1 learning process using well-established credit-assignment or temporal-difference methods. It is better to justify when or why a hierarchical design offers significant advantages over this simpler alternative.

(3) The current work focuses solely on the offline (off-policy) setting, assuming access to logged data and sidestepping challenges related to exploration in dynamic environments. In practical deployments, online learning systems must balance exploration and exploitation, especially when optimizing long-term outcomes. It remains unclear how the proposed framework can be extended to the online setting or whether it is compatible with well-known exploration techniques such as UCB or Thompson sampling.

(4) The related work section omits several relevant and recent contributions from the neural contextual bandit community, which tackle closely related problems such as exploration, long-term reward optimization, and hierarchical modeling. Notable examples include:
	•	Neural contextual bandits with UCB-based exploration
	•	EE-Net: Exploitation–Exploration Neural Networks in contextual bandits
	•	Graph Neural Network Bandits

---

> ### Author Rebuttal · Authors · 2025-07-31
>
> We thank the reviewer for their time and effort for this valuable service.
>
> **We are glad that all four reviewers unanimously acknowledge the novelty of the method with high scores along the axis of Originality**.
> We are glad that the reviewer found our proposed algorithm "easy to plug into existing ranking, LLM decoding or recommendation stacks" and noted that "Across chat, simulated and real-world recommender tasks, MSBL consistently lifts long-horizon objectives while keeping click/engagement drops small and showing resilience to noisy features or larger action spaces."
>
> The concerns raised are primarily misunderstandings. We clarify each of them and address them below.
>
> > Empirical support for core motivation
>
> - In recommender systems, over-optimization of clicks leading to lower user retention [1, 2], challenges in aligning short, medium and long session goals [3] and challenges with competing time horizon objectives [4] have been extensively reported. For instance, [4] describes time horizon objectives in Figure 1 (short term as click through rate, and long term as customer satisfaction) and states that “long-term and short-term objectives may be competing”. They discuss existing recommendation paradigms and conclude that “Such strategies that are successful in the short term may however be non-optimal or even detrimental in the long run.”
>
> - This misalignment between micro and macro objectives has been shown to be substantial [2,8]. Specifcially, Figure 2 of [2] shows the dcg metric for the micro/short term utility of micro actions (ranking) and the macro violation for the long term constraint violation. Similarly, [8] shows the tradeoff between utility and amortized exposure (collected over a time horizon T) as the macro goal. These works assume that the macro intervention is given while our work learns it via the macro policy.
>
>
> - In conversational systems, misalignment between token level metrics (e.g., perplexity) and multiple responses/document level metric (e.g., diversity in a paragraph) has been studied both qantitatively and qualitatively in [5, 6, 7].
>
> As a result, there is overwhelming evidence that short-term objectives can be substantially misaligned with long-term objectives across a wide range of applications.
>
> Please also note that most reviewers found the problem important and timely. Reviewer mEmM noted “The problem of optimizing long-term objectives in AI systems is both timely and important, especially given the prevalence of short-term optimization in practice.”, reviewer PY8n noted "The paper addresses a critical challenge in interactive AI systems, where short-term feedback is abundant, but optimizing solely for it often harms long-term objectives." and reviewer enko also noted “The problem being addressed is ubiquitous in practical applications, and so quite important.”
>
> > short-term metrics often correlate well with long-term outcomes, which the paper itself implicitly assumes when using L1 feedback to construct priors for L2 learning.
>
> This is an important and subtle point. While short term metrics provide an important signal for long term outcomes, their unmitigated optimization is not necessarily aligned with long term goals (as we provided evidence above).
> In our work, we provide a principled way of modifying the policy or the feedback so that while there may be a sacrifice in the short term metrics (e.g clicks, perplexity of a sentence) but the modified policy is more effective at optimizing upper level reward (e.g., fewer clicks by more aggressively pruning suspected click bait can lead to better weekly returns or transforming the short term rewards to steer for long term goals). See lines 189-192 and lines 206-208 in the main text.
>
> > when/why our method provides advantage over existing flat learning methods
>
> - Existing flat learning methods such as [2,9] consider optimizing for L1 actions only (rankings). They do not learn L2 interventions such as reward weights [10], ads [11], ad pacing [12], message notifications, decoding strategies [5] etc that are available in the system and can directly steer the system towards long term objectives.
> - The literature on delayed rewards such as [9] and the related works that the reviewer pointed out model the delayed reward of a micro level action. Crucially, they do not address the situation where the delayed long term reward cannot be attributed to a single micro level action, but is the result of multiple actions. For instance, diversity among T generated responses by an LLM policy depends on all the responses generated within the T horizon; similarly, a specific exposure target as a long term goal depends on rankings over the entire T horizon [2].
> - unlike the flat learning methods [2,9], our framework extends to more than two timescales of competing objectives.
> These flat learning methods can in fact be plugged into a single level (usually the lowest level) in the MSPL framework.
>
>
> > Focus on off-policy
>
> In this work we focus on off policy learning (OPL) using naturally collected logged bandit data.
> - OPL is of great practical relevance, as it enables improving the system without the risky, slow, and potentially unethical use of online exploration [13,14,15].
> - Importantly, **access to logged bandit data is not considered an assumption but is naturally collected in most real setting [13,14,15]** .
> - While one could swap out the off-policy method with an on-policy method (e.g., REINFORCE policy gradient) and develop MSPL with online estimators, it is out of the scope of the current work. Instead we discuss the deployment of MSBL policies and their adaptation at inference time (as new data arrives).
>
>
> > Adapting to dynamic environments
>
> Please refer to the response to reviewer mEmM on “Updating MSBL policies as new data arrives”. We provide clarification on how the different levels can be updated asynchronously (offline) and provide empirical results.
>
> > Exploration-exploitation tradeoffs in online systems
>
> Addressing exploration-exploitation tradeoffs is important but orthogonal to the focus of our work.
> We described in the response under “Updating MSBL policies as new data arrives” how policies at different levels can be updated asynchronously. With that discussion as the context, one can also update MSBL policies incorporating exploration from existing UCB or thompson sampling works to balance this tradeoff.
> To reiterate, centering our work on off-policy learning (OPL) has its own merits and practical relevance [15].
>
> > Related works
>
> We will move the discussion on related works in this response and that in the appendix to the main text to incorporate these clarifications.
>
> [1] Focusing on the Long-term: It's Good for Users and Business (Hohnhold et al'15)
>
> [2] Ranking with Long term constraints (Brantley et al'24)
>
> [3] A Survey on Session-based Recommender Systems (Wang et al'21)
>
> [4] A survey on multi-objective recommender systems (Jannach et al'23)
>
> [5] Adaptive Decoding via Latent Preference Optimization (Dhiliawala et al'24)
>
> [6] Perplexity by PLM Is Unreliable for Evaluating Text Quality (Wang et al'23)
>
> [7] Escaping the sentence-level paradigm in machine translation (Post et al'24)
>
> [8] Controlling Fairness and Bias in Dynamic Learning-to-Rank (Morik et al'20)
>
> [9] Impatient Bandits (Zhang et al'24)
>
> [10] From Optimizing Engagement to Measuring Value (Milli et al'21)
>
> [11] Control-based Bidding for Mobile Livestreaming Ads with Exposure Guarantee (Zhang et al'22)
>
> [12] Smart Pacing for Effective Online Ad Campaign Optimization (Xu et al'15)
>
> [13] Counterfactual Risk Minimization: Learning from Logged Bandit Feedback (Swaminathan et al'15)
>
> [14] Off-Policy Evaluation for Large Action Spaces via Policy Convolution (Sachdeva et al'23)
>
> [15] Off-Policy Evaluation and Learning in Recommender Systems (Recsys 2021 Tutorial)

---

> > ### Comment · Reviewer_z6c1 · 2025-08-04
> >
> > Thank you for the authors’ detailed response. It addresses the majority of my concerns. Please incorporate these new clarifications into the final manuscript. I will increase the score from 3 to 4.

---

> > > ### Author Response · Authors · 2025-08-04
> > >
> > > We thank the reviewer for their feedback and for positive evaluation of the paper. We will incorporate these clarifications in the final manuscript. Please let us know if we can address anything else to further improve your evaluation of the paper.

---

### Official Review · Reviewer_PY8n · 2025-07-03

**Clarity:** 2
**Significance:** 2
**Originality:** 3
**Rating:** 4
**Confidence:** 3

**Summary:**

The paper addresses a critical challenge in interactive AI systems, where short-term feedback is abundant, but optimizing solely for it often harms long-term objectives. The authors propose the MultiScale Policy Learning framework to reconcile the difference in feedback timescales. Specifically, they present MSBL, which leverages plentiful short-term data to establish hierarchical priors for faster learning at higher, less data-rich levels. Empirical evaluations demonstrate performance improvements on tasks related to recommender and conversational systems.

**Questions:**

See weakness.

**Ethical Concerns:**

["NO or VERY MINOR ethics concerns only"]

**Final Justification:**

Thanks for the author's detailed response. My concerns are addressed, and I will keep the positive score.

**Limitations:**

Yes

**Quality:**

2

**Strengths And Weaknesses:**

Strengths:

(1) Effectively integrates feedback across multiple timescales, ensuring better alignment between short-term actions and long-term objectives.

(2) Uses abundant short-term feedback to inform hierarchical priors, dramatically enhancing data efficiency for learning sparse, long-term objectives.

Weakness:

(1) This method heavily relies on accurate short-term feedback to construct hierarchical priors for long-term optimization. If the short-term feedback is biased or noisy, will the newly formed prior adversely affect the long-term optimization? If yes, does this significantly reduce its advantages compared to directly learning from short-term feedback?

(2) Implementing and optimizing multi-level hierarchical policies introduces substantial computational complexity. What is the additional computational cost incurred compared to conventional single-level feedback learning? Is this increased cost justified by the performance improvements?

---

> ### Author Rebuttal · Authors · 2025-07-31
>
> We thank the reviewer for their time and effort for this valuable service.
>
> **We are glad that all four reviewers unanimously acknowledge the novelty of the method with high scores along the axis of Originality**.
>
> The concerns raised are primarily clarifications. We address each of them below.
>
> > How does errors/noise in short term feedback affect macro policy
>
> Our experiment in Figure 13 of Appendix looks at how noise in micro policy affects macro policy learning. We evaluate for varying levels of noise by perturbing the ranking scores.
> While the noise at micro level does affect long term optimization, we find that empirically macro learning is still robust to fairly large values of noise at micro level and maintains its advantage over the baselines (including the baseline of only learning from short term "short term opt").
> Please also see the response to reviewer mEmM under “how much the prior advantage contributes to the macro-level improvement”.
>
> > Computational cost of hierarchical learning
> - In order to compare the computation cost we analyze the samples required for directly optimizing long term feedback versus our approach of hierarchical learning in section 3. While MSPL learns a policy at each level, eq 4 shows that hierarchical learning would be beneficial to learning a single- level policy from an uniformed prior depending on the misalignment between different timescale feedback. As reviewer  z6c1 noted “By re-using micro-level knowledge as a Bayesian prior, MSBL needs dramatically fewer long-term samples to reach reliable macro policies—crucial when retention or renewal labels are rare.”
> - Aditionally, as noted in lines 234-236,  the macro action space (decoding strategies, reward weights etc.) is much smaller than that of the micro level (rankings, token action space) so that the macro policies are progressively smaller in size.

---

> > ### Comment · Reviewer_PY8n · 2025-08-05
> >
> > Thanks for the author's detailed reponse. My concerns are addressed and will keep my postive score.

---

### Official Review · Reviewer_mEmM · 2025-07-03

**Clarity:** 3
**Significance:** 2
**Originality:** 3
**Rating:** 4
**Confidence:** 3

**Summary:**

This paper introduces a novel MultiScale Policy Learning framework for contextual bandits aimed at optimizing long-term objectives in interactive AI systems. The key insight is to address the mismatch between short-term interventions (e.g., clicks) and long-term outcomes (e.g., user retention) by learning policies across multiple interdependent timescales. The proposed MultiScale Off-Policy Bandit Learning (MSBL) leverages abundant short-term feedback to construct informative priors for higher-level policies, enabling more sample-efficient learning from sparse long-term signals. The framework is instantiated and evaluated on simulated tasks involving two to three levels of decision-making in both recommendation and conversational system scenarios.

**Questions:**

1) Is there a way to update the macro policies online as new macro-level data arrives, or is the learning strictly offline?
2) Can the proposed multi-scale framework be extended to reinforcement learning beyond bandits?
3) While the PAC-Bayes provides useful insight, it’s not empirically tested how much the prior advantage contributes to the macro-level improvement.

**Ethical Concerns:**

["NO or VERY MINOR ethics concerns only"]

**Final Justification:**

Most of my concerns regarding the empirical performance have been addressed thus I will maintain my score.

**Limitations:**

Yes.

**Quality:**

3

**Strengths And Weaknesses:**

Strengths
1) The problem of optimizing long-term objectives in AI systems is both timely and important, especially given the prevalence of short-term optimization in practice.
2) The hierarchical treatment of feedback across multiple timescales is novel, providing a compelling approach to reconciling short-term actions with long-term outcomes.
3) The experimental evaluation is thorough and well-structured, spanning diverse domains such as recommendation and conversational systems.

Weaknesses
1) While the PAC-Bayesian motivation provides valuable theoretical insight, the analysis remains somewhat informal. A more rigorous treatment—such as formal guarantees or generalization bounds for the full multi-level case—would strengthen the theoretical contribution.
2) The experimental setup, though thoughtfully designed, is primarily based on synthetic simulators. While this is reasonable for initial validation, the absence of real-world datasets or deployment severely limits the practical applicability and external validity of the results.
3) The framework presumes the existence of feedback clearly aligned to multiple levels (e.g., clicks, weekly returns, subscriptions), which may not be available or well-defined in some domains.

---

> ### Author Rebuttal · Authors · 2025-07-31
>
> We thank the reviewer for their time and effort for this valuable service.
>
> **We are glad that all four reviewers unanimously acknowledge the novelty of the method with high scores along the axis of Originality**. To quote, the reviewer  noted “The hierarchical treatment of feedback across multiple timescales is novel, providing a compelling approach to reconciling short-term actions with long-term outcomes.” Reviewer enko also noted “The proposed hierarchical separation is quite general and powerful in capturing a wide range of situations.”
>
> The concerns raised are primarily clarifications. We address each of them below.
>
> > experiment setup on synthetic simulators
>
> The reviewers largely agree that the paper tackles a significant, timely and an important problem.
> To progress research in this direction, we used real world datasets/ setup in existing papers [1,4,5] for level 1 and simulate medium and long term feedback (following previous works [2,3,6]). For our simulator with three levels, we describe the design and modules in Figure 6 of Appendix and open source it as part of the paper.<!-- These are semi-synthetic experiments and the best available option in the absence of open source real world datasets for this problem.  -->
> Please also refer to our response to reviewer enko under "empirical evaluation".
>
> [1] Rewards-in-Context: Multi-objective Alignment of Foundation Models with Dynamic Preference Adjustment (Yang et al'24)
>
> [2] Ranking with Long term constraints (Brantley et al'24)
>
> [3] Regressing the Relative Future: Efficient Policy Optimization for Multi-turn RLHF (Gao et al'25)
>
> [4] Anthropic’s helpful-harmless dataset
>
> [5] KuaiRec dataset for recommender systems (Gao et al'22)
>
> [6] LLM as judge
>
> > PAC Bayesian theoretical insights vs guarantees
>
> Thanks for acknowledging the valuable theoretical insight of the PAC Bayes motivation. While we provide an upper bound for improvement in required training samples in eq (4), providing a practically meaningful formal guarantee is challenging given the complexity of the models involved. We see the value of the PAC Bayes analysis in validating the principle behind our model, not quantifying the outcome.
>
> > Updating MSBL policies as new data arrives
>
> This is a great question. The macro policy is adaptable and can be updated when new data arrives. An on policy estimator (e.g., REINFORCE policy gradient) could be used but our work is centered on off-policy learning (OPL) setting. Below we describe the scenarios and provide empirical results with and without updating L2 policy (offline) as new macro data arrives.
>
> MSPL decouples the micro and macro policy learning so that they can be updated asynchronously.
> - If new macro level data arrives (e.g changing user preferences), the family of policies at the micro level do not need to be updated. We only need to update the macro policy that selects the optimal macro intervention according to the new macro data.
> This is advantageous as compared to methods that reviewer z6c1 referred to as incorporating L2 learning as part of L1 learning policy. For such methods, one would need to update a single monolithic deployed policy if any new data arrives. MSPL's modular design allows decoupled learning between levels.
>
>     To demonstrate this empirically, we conduct an experiment for the last setup - two levels of large scale recommendation system with the MSBL policies deployed. We simulate a scenario such that the underlying user preferences change after deployment. In the below results the first row "w. new macro data" shows both long and short term reward without updating the L2 policy and the second row shows the expected rewards after L2 policy is updated according to the new macro level data.
>     Across different number of user groups we can see that while the micro level (clicks) remains mostly unchanged, **only** updating macro policy results in lifting the long term.
>     | 2 Groups | Long term (Return Rate) $\uparrow$ | Short term (Clicks) $\uparrow$|
>     |----------|----------|----------|
>     | w. new macro data    | 0.64$\pm$ 0.01   | 239.62 $\pm$ 0.68    |
>     | w. L2 policy updated    | 0.65 $\pm$ 0.01    | 239.70 $\pm$   0.62 |
>     ||||
>     | % Improv. w. updating L2    | $\mathbf{1.37}$ %  | $0.03$ %   |
>
>     | 3 Groups | Long term (Return Rate) $\uparrow$ | Short term (Clicks) $\uparrow$|
>     |----------|----------|----------|
>     | w. new macro data    | 0.48 $\pm$ 0.12   | 240.50 $\pm$ 0.76    |
>     | w. L2 policy updated    | 0.54 $\pm$ 0.07    | 240.13 $\pm$   0.41 |
>     ||||
>     | % Improv. w. updating L2    | $\mathbf{14.46}$ %  | $-0.15$ %   |
>
>     | 4 Groups | Long term (Return Rate) $\uparrow$ | Short term (Clicks) $\uparrow$|
>     |----------|----------|----------|
>     | w. new macro data    | 0.42 $\pm$ 0.08   | 240.48 $\pm$ 0.78    |
>     | w. L2 policy updated    | 0.45 $\pm$ 0.06    | 240.42 $\pm$   0.83 |
>     ||||
>     | % Improv. w. updating L2    | $\mathbf{8.14}$ %  | $-0.02$%   |
> - On the other hand, new micro-level data (for instance with addition of items in catalog, updates required in the base LLM policy etc.) would require updating both micro and macro policy.
>
> - Finally, this framework elegantly scales to more than two levels when new data arrives. For instance, for three levels, if new data arrived for L2, only L2 and L3 policy need to be updated. The L1 policy would remain unchanged. Below, we show with an experiment on the second setup - a three level simulator that new L2,L3 data only affects the medium and long term and the L1 policy does not need to be updated.
>
>     | 2 Groups | Long term | Medium term  |Short term  |
>     |----------|----------|----------|----------|
>     | w. new L2, L3 data    | 0.88   | 0.85    |0.52   |
>     | w. L2, L3 policy updated    | 0.98   | 0.90   |0.52    |
>     |||||
>     | % Improv. w. updating L2 & L3   | $\mathbf{11.59 }$ % | $\mathbf{5.76}$ %  | $0.61$%  |
> > Extending to reinforcement learning beyond bandits
>
> This is another great point. We note in limitations, line 354 that “extension to stateful policies would be an interesting future work”. We conjecture such an extension to be straightforward especially if dependencies at the lower level stay within the timestep of the next higher level. For example, in experiment 2 (conversational recommender simulator) for the lowest level, we already append the response of the L1 LLM policy at each timestep to the next context, so that the current context depends on the action taken in previous timestep.
> However, a more general and fully principled extension of MSPL to MDP policies requires careful consideration and challenges related to learning MDP policies with scarce data.
>
> Regarding
> > how much the prior advantage contributes to the macro-level improvement,
>
> To evaluate the prior advantage from micro level learning, we conduct an experiment where we inject noise into the micro learning process and see how it affects macro level rewards for the last setup - two levels of large scale recommendation system. A noise of 0.4 refers to 40% randomness in micro action selection.
> Below , we can see that the advantage of micro level when learned (w. 0.0 noise) is significant compared to a random micro policy (w. 1.0 noise)
>
> |Noise in Micro Policy | Long term (Return Rate) $\uparrow$|
> |----------|----------|
> | 0.0   | $\mathbf{0.82 \pm 0.01}$|
> | 0.4|0.81 $\pm$ 0.01 |
> | 0.6|0.79 $\pm$ 0.01 |
> | 1.0 (random micro policy)    | $\mathbf{0.72 \pm 0.01}$ |

---

> > ### Author Response · Authors · 2025-08-05
> >
> > Could the reviewer let us know whether our response has clarified the raised concerns, and whether this changes the reviewers evaluation of the paper.

---

> > ### Comment · Reviewer_mEmM · 2025-08-06
> >
> > Thank you for the detailed response. Most of my concerns regarding the empirical performance have been addressed, and I will maintain my current score.

---

### Note · Authors · 2025-08-12

We thank the reviewers for their thoughtful feedback and positive assessment.
According to the reviewers, our work provides a novel and compelling approach to effectively address the disconnect between short term actions and long term objectives, a problem that is important and ubiquitous in a variety of domains.
Specifically, we provide a PAC Bayes motivation to propose a policy learning framework at multiple timescales. We instantiate this framework with off policy contextual bandits (MSBL) and provide two ways - policy and feedback modification that enable the use of both abundant short-term and sparse long-term data for optimizing long-term outcomes. Across chat, simulated and real-world recommender tasks, MSBL consistently lifts long-horizon objectives while keeping click/engagement drops small and showing resilience to noisy features or larger action spaces.

During the rebuttal phase, we provided detailed responses addressing the following concerns,
1. evidence for selecting the empirical tasks and baselines in line with existing related literature.
2. how to update MSBL policies as new data arrives (clarification and empirical results)
3. effect of noise in micro policy on macro performance and vice versa (clarification and empirical results)
4. Further clarifications on minor concerns such as difference from flat learning methods, factorization of policies, and other technical details.

Since Neurips guidelines specifically state to not pay attention to score values but the review text, we did not focus on the numerical score rather engaged with reviewers on addressing their concerns.
All reviewers responded that their key concerns have been addressed. We also asked the reviewers for any additional concern that they have and that we could address during the discussion period.
Reviewer z6c1 responded that they will increase their score and reviewer enko responded that they will update the score accordingly.

---

### Decision · Program_Chairs · 2025-09-17

**Decision:**

Accept (poster)

**Comment:**

In the space of interactive/online learning, the paper investigates the balance between optimizing for abundant, short-term feedback (e.g., clicks, engagement) and achieving desired long-term objectives (e.g., user retention, satisfaction). The authors identify the different timescales of interventions and feedback as a key obstacle and as a solution introduce the MultiScale Policy Learning (MSPL) framework. The framework leverages abundant short-term feedback to form hierarchical priors that accelerate learning from sparse long-term signals.  A PAC-Bayesian motivation is provided to show why such hierarchical priors lead to data efficiency. Empirical evaluations on simulated conversational tasks and recommender systems demonstrate consistent improvements on long-term metrics, with minimal sacrifice of short-term performance. The reviewers were leaning positive towards the paper and noted that the paper is timely and novel, offering a lightweight wrapper around standard bandits and demonstrating consistent gains on conversational and recommender tasks. However, reviewers also raised important concerns about reliance on simulators, robustness to noise, offline nature of the framework, lack of rigorous theoretical guarantees, and limited quantitative evidence for the claimed timescale misalignment. We thank the authors and the reviewers for engaging during the rebuttal period towards improving the paper and providing additional experiments (online policy updates and robustness to noisy micro policies). The clarification in rebuttal along with new results resolved many of the reviewers concerns, with some raising their scores. Overall, while limitations remain in theory and real-world validation, the work presents a compelling and original contribution and thus I lean recommending acceptance to NeurIPS given the significance of the problem and positive reviewer consensus.